# Epithelial colonies in vitro elongate through collective effects

Jordi Comelles[1,2,3,4], Soumya SS[5], Linjie Lu[1,2,3,4], Emilie Le Maout[1,2,3,4], S Anvitha[6], Guillaume Salbreux[7], Frank Jülicher[8,9], Mandar M Inamdar[5]*, Daniel Riveline[1,2,3,4]*

[1]Laboratory of Cell Physics ISIS/IGBMC, CNRS and Université de Strasbourg, Strasbourg, France; [2]Institut de Génétique et de Biologie Moléculaire et Cellulaire, Illkirch, France; [3]Centre National de la Recherche Scientifique, UMR7104, Illkirch, France; [4]Institut National de la Santé et de la Recherche Médicale, U964, Illkirch, France; [5]Department of Civil Engineering, Indian Institute of Technology Bombay, Powai, Mumbai, India; [6]Department of Mechanical Engineering, Indian Institute of Technology Bombay, Powai, Mumbai, India; [7]The Francis Crick Institute, London, United Kingdom; [8]Max Planck Institute for the Physics of Complex Systems, Dresden, Germany; [9]Cluster of Excellence Physics of Life, Dresden, Germany

**Abstract** Epithelial tissues of the developing embryos elongate by different mechanisms, such as neighbor exchange, cell elongation, and oriented cell division. Since autonomous tissue self-organization is influenced by external cues such as morphogen gradients or neighboring tissues, it is difficult to distinguish intrinsic from directed tissue behavior. The mesoscopic processes leading to the different mechanisms remain elusive. Here, we study the spontaneous elongation behavior of spreading circular epithelial colonies in vitro. By quantifying deformation kinematics at multiple scales, we report that global elongation happens primarily due to cell elongations, and its direction correlates with the anisotropy of the average cell elongation. By imposing an external time-periodic stretch, the axis of this global symmetry breaking can be modified and elongation occurs primarily due to orientated neighbor exchange. These different behaviors are confirmed using a vertex model for collective cell behavior, providing a framework for understanding autonomous tissue elongation and its origins.

*For correspondence:
minamdar@iitb.ac.in (MMI);
riveline@unistra.fr (DR)

**Competing interests:** The authors declare that no competing interests exist.

## Introduction

Tissue elongation is a central morphogenetic event occurring in many organisms during development (*Lecuit and Lenne, 2007*; *Guillot and Lecuit, 2013*), such as *Drosophila* or *Caenorhabditis elegans*. The tissue is transformed both in terms of area and shape. Such transformation takes place within typically hour timescale with or without cell division. During this process, symmetry of cells and tissues is broken by different mechanisms, such as neighbor exchange (*Rauzi et al., 2008*; *Rauzi et al., 2010*), cell elongation (*Ciarletta et al., 2009*; *Vuong-Brender, 2016*), and oriented cell division (*Campinho et al., 2013*). Rearrangement of neighboring cells or T1 transitions is essential in the germ band extension of *Drosophila* (*Rauzi et al., 2008*; *Rauzi et al., 2010*), allowing a group of cells to change their position by intercalation, eventually leading to tissue elongation. Cell deformation drives the threefold elongation process in *C. elegans* (*Ciarletta et al., 2009*; *Vuong-Brender, 2016*) while keeping the number of cells and their positions fixed. Finally, epithelial spreading during zebrafish epiboly is promoted by oriented cell divisions as a mechanism to limit tension (*Campinho et al., 2013*). Those mechanisms can act alone or in combination as in *Drosophila* pupal wing elongation (*Etournay et al., 2015*). While the phenomenon is known to involve remodeling of adherens junctions (*Rauzi et al., 2010*) and acto-myosin (*He et al., 2010*; *Rauzi et al., 2010*) at the molecular level, mesoscopic mechanisms leading to distinct morphogenesis processes are poorly

understood. This is partly because inputs from morphogen gradients (*Gilmour et al., 2017*) or from neighboring tissues (*Zhang et al., 2011*; *Etournay et al., 2015*) can affect tissue autonomous self-organization in vivo. For example, changes in tissue shape can be influenced by neighboring tissues such as the cuticle and the blade in the *Drosophila* pupal wing elongation (*Etournay et al., 2015*), the coordination between amnioserosa and epidermis in dorsal closure (*Hayes and Solon, 2017*), and the muscle layer in gut development (*Shyer et al., 2013*) or in *C. elegans* morphogenesis (*Zhang et al., 2011*). Since in vivo, epithelial tissues are surrounded by other tissues and the cellular dynamics leading to elongation can result from interactions between tissues and boundary conditions, it is therefore difficult to disentangle cell intrinsic from externally mediated behaviors. In this context, it appears important to characterize elongation in an in vitro system where the epithelial tissue undergoes shape transition autonomously.

Here, we use in vitro experiments and numerical simulations to characterize the spontaneous behavior of a growing cell colony in vitro. We designed an assay to study the spontaneous elongation of a tissue that is not subjected to external orienting input, we studied the appearance of the symmetry breaking, and the effect that external forces have in this process. We show that in vitro tissue elongation arises from anisotropy in the average cell elongation. This anisotropy sets the direction along which boundary cells migrate radially outwards resulting in a non-isotropic elongation that arises primarily through cell elongation. For colonies submitted to a time periodic uniaxial stretch, the axis of global symmetry breaking can be imposed by external force, and tissue elongation arises through oriented neighbor exchange. Emergence of radially migrating cells and the interplay between cell elongation and cell rearrangements are confirmed by numerical simulations based on a vertex model. Our results suggest that spontaneous shape deformation is related to the mean orientation of the nematic cell elongation field in the absence of any external input. This provides a framework to explain autonomous tissue elongation and how contributions from different mesoscopic mechanisms can be modulated by external forces.

## Results

### Isotropic colonies elongate in a non-isotropic manner

To study the spontaneous tissue deformation arising during epithelial growth, we designed an in vitro assay to track symmetry breaking, both spontaneous and driven by external force. We prepared isotropic colonies of Madin Darby Canine Kidney (MDCK) cells, which assume features of epithelial cells in vivo (*Reinsch and Karsenti, 1994*; *Adams et al., 1998*; *Reffay et al., 2014*), such as adherens junctions (*Adams et al., 1998*), cytoskeletal components, and the Rho signaling pathway regulating cell shapes and dynamics (*Reffay et al., 2014*; *Fodor et al., 2018*). The initial size and shape of the colonies were controlled by plating cells in microfabricated circular stencils (*Ostuni et al., 2000*). When cells reached confluency, the stencil was removed at time $t_0$. Cell dynamics was followed over time by phase contrast (*Video 1*) or fluorescence microscopy with strains labeled with GFP cadherin (*Figure 1b*), that allowed to observe the behavior of individual cells. We observed that large colonies (750 µm in diameter) expanded isotropically (*Figure 1—figure supplement 1*). In contrast, colonies of 250 µm in diameter (*Figure 1b*), the typical coherence length of such epithelial tissues (*Doxzen et al., 2013*), expanded in a non-isotropic manner (*Figure 1c*).

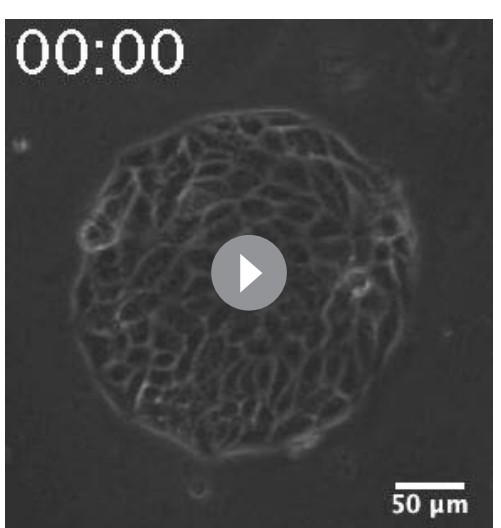

**Video 1.** Spontaneous symmetry breaking of circular colonies. Time-lapse of a Madin Darby Canine Kidney (MDCK) cells colony freely evolving after removal of a poly(dimethylsiloxane) (PDMS) stencil. Time in hh:mm. Scale bar 50 µm.
https://elifesciences.org/articles/57730#video1

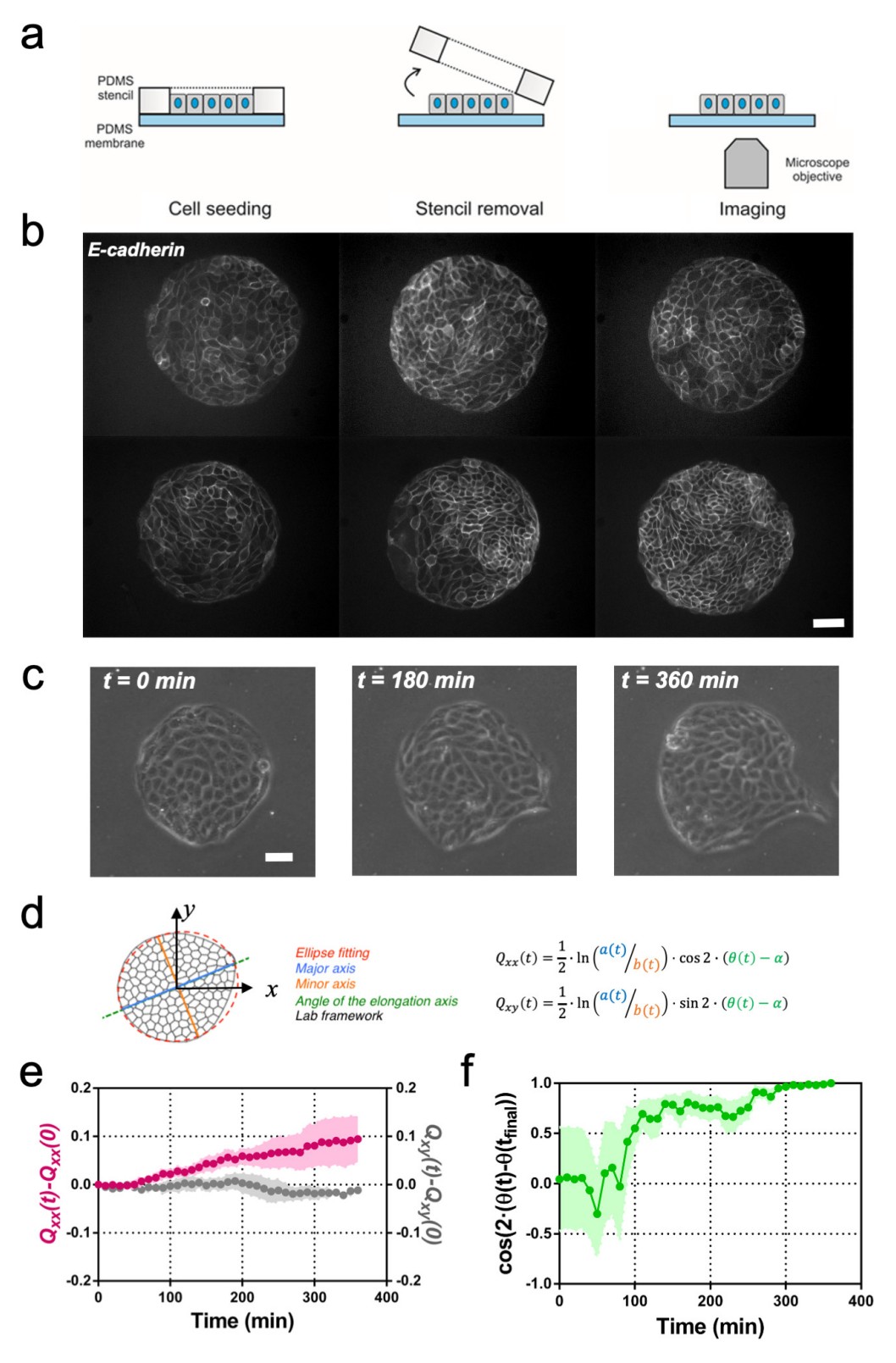

**Figure 1.** Symmetry breaking and its quantification. (**a**) Schematics of the experimental set-up: Madin Darby Canine Kidney (MDCK) cells were seeded on a poly(dimethylsiloxane) (PDMS) membrane using stencils to predefine their shape. When the colony was confluent, the stencil was removed and the expansion of the colony was observed under the microscope. (**b**) Several examples of MDCK colonies (GFP-E-cadherin) after stencil removal and prior to colony expansion. Scale bar 50 μm. (**c**) Phase contrast images of the spontaneous elongation of an MDCK colony for 360 min. Scale bar 50 μm. (**d**)

*Figure 1 continued on next page*

*Figure 1 continued*

Colony elongation is quantified by ellipse fitting and $Q_{xx}$ and $Q_{xy}$ measurement referred to the elongation axis ($\alpha = \theta(t_{final})$). (**e**) $Q_{xx}$ (left y axis) and $Q_{xy}$ (right y axis) during 360 min of colony expansion. Mean value ± standard error of the mean, n = 4 colonies from N = 4 independent experiments. (**f**) Cosine of two times the angle difference between the instantaneous main direction of the colony ($\theta(t)$) and the main direction of the colony at 360 min ($\theta(t_{final})$). Colonies set the elongation direction within the first 120 min. Mean value ± standard error of the mean, n = 4 colonies from N = 4 independent experiments.

The online version of this article includes the following source data and figure supplement(s) for figure 1:

**Source data 1.** The $Q_{xx}(t)$-$Q_{xx}(0)$ and $Q_{xy}(t)$-$Q_{xy}(0)$ of individual colonies used in panel (e) and the raw values of $\cos(2 \cdot (\theta(t)-\theta(t_{final})))$ used in panel (f) of *Figure 1*.

**Figure supplement 1.** Madin Darby Canine Kidney (MDCK) colonies of 750 µm in diameter expand isotropically.

**Figure supplement 1—source data 1.** $Q_{xx}(t)$-$Q_{xx}(0)$ and $Q_{xy}(t)$-$Q_{xy}(0)$ for the three 750 µm in diameter individual colonies of MDCK cells expanding for 6 hours.

**Figure supplement 2.** MCF 10A and Caco-2 colonies of 250 µm in diameter expand asymmetrically similar to Madin Darby Canine Kidney (MDCK) colonies.

**Figure supplement 2—source data 1.** Raw data corresponding to panels (c) and (d): $Q_{xx}(t)$-$Q_{xx}(0)$ and $Q_{xy}(t)$-$Q_{xy}(0)$ for individual colonies (250 µm in diameter) of MCF 10A cells and Caco-2 expanding for 6 hours.

We then further characterized the process of symmetry breaking.

In order to compare elongations in each experiment, we quantified the breaking of symmetry by ellipse-fitting the colony shape. Shape change analysis was quantified by a nematic shape elongation tensor **Q**. It has two independent components defined as $Q_{xx} = \frac{1}{2} \ln(a/b)\cos(2\bullet(\theta - \alpha))$ and $Q_{xy} = \frac{1}{2} \ln(a/b)\sin(2\bullet(\theta - \alpha))$, where *a* corresponds to the major axis, *b* to the minor axis, $\theta$ to the orientation of the major axis of the ellipse and $\alpha = \theta(t_{final})$ (*Figure 1d*). As can be seen in *Figure 1e*, MDCK colonies elongated persistently along the main axis of elongation ($Q_{xx} > 0$ and $Q_{xy} \approx 0$) for 6 hr (*Figure 1e*). In addition, we explored if other epithelial cell lines would behave in a similar manner. Circular epithelial colonies of human epithelial colorectal adenocarcinoma cells (Caco2) and human mammary epithelial cells (MCF-10A) also elongated along the main axis of elongation and by the same magnitude that MDCK cells (*Figure 1—figure supplement 2*). We note that elongation observed during this time for the three epithelial cell lines was similar in magnitude to tissue elongation observed during in vivo morphogenesis, for instance in the wing blade in *Drosophila* (*Etournay et al., 2015*). Moreover, the elongation direction ($\theta_{final} = \theta (t = 6 h)$) converges to a constant value within 2 hr after $t_0$ (*Figure 1f*). Altogether, large circular epithelial colonies (750 µm in diameter) expand isotropically, whereas small colonies (250 µm in diameter) expand in a anisotropic manner and shape symmetry breaking takes place within the first 2 hr. As a result, we focus here on these first 2 hr during which the elongation axis is established.

## Cyclic uniaxial stretching rectifies symmetry breaking

It has been previously described for *C. elegans* embryo elongation (*Zhang et al., 2011*) and in other organisms (*Zhang and Labouesse, 2012*) that time periodic stretch can play a role in morphogenesis. Motivated by these observations, we explored whether oscillatory external forces could have an impact on the direction of elongation. We designed an experimental setup where elongating colonies were submitted to cyclic uniaxial stretching (*Figure 2a* and *Video 2*). Mechanical cycles of contraction-relaxation can range from 1 s in *C. elegans* epithelial elongation (*Zhang et al., 2011*) up to 200 s in dorsal closure (*Solon et al., 2009*). So, we explored frequencies and extensions around physiological values (*Zhang and Labouesse, 2012*). We selected three different cycle durations (20, 60, and 120 s period) and three different stretching conditions (5%, 10% and 15% strain). The stretch was applied to a silicon membrane and was transmitted to the colony. We then fitted the colonies shapes with ellipses at successive time and quantified $Q_{xx}$ and $Q_{xy}$ with respect to the angle of uniaxial stretching (set as *x*-axis, $\alpha = 0$). *Figure 2—figure supplement 1* shows the value of the components of the tensor **Q** along time for the different strains and periods tested. Among different conditions, we observed colony elongation along the direction imposed by the external strain when we stretched cyclically with 60 s timescale and 5% strain (*Figure 2b* and *Video 3*). The overall elongation of colonies under cyclic uniaxial stretching was similar to the spontaneous elongation in the absence of externally applied uniaxial stretching during the first 2 hr (*Figure 2c*). Also, the magnitude of the shape elongation tensor **Q** under cyclic uniaxial stretching was comparable to the

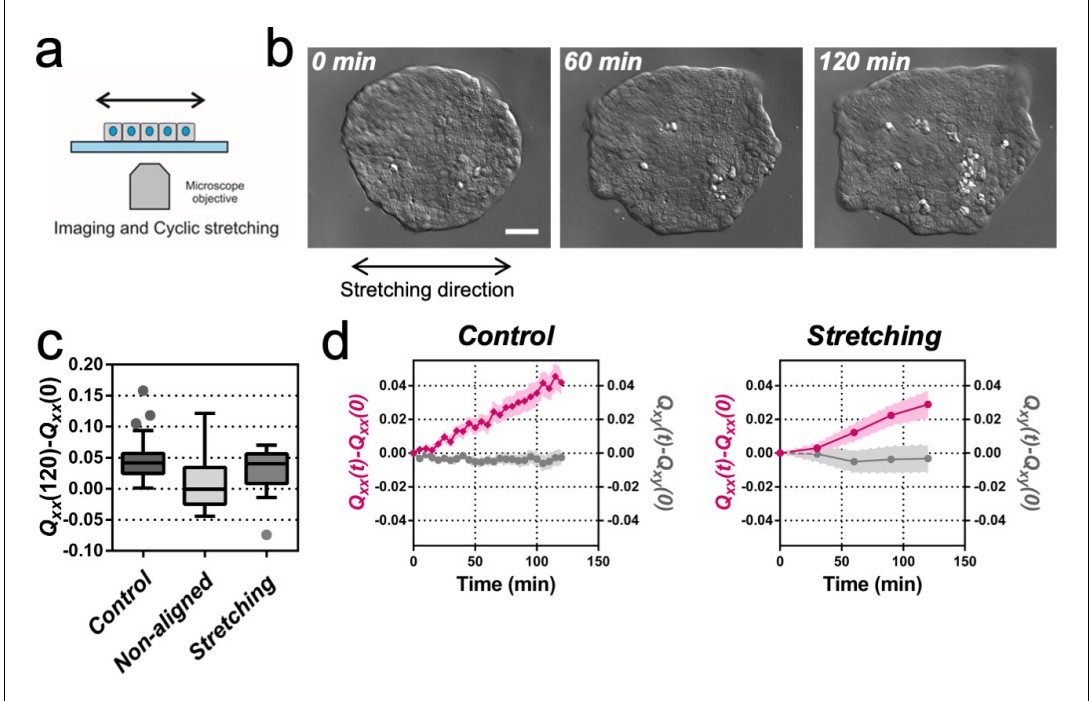

**Figure 2.** Uniaxial cyclic stretching rectifies symmetry breaking. (a) Schematics of the experiment where the colony expansion was observed under the microscope while the underlying membrane was uniaxially stretched. (b) Snapshots of the expansion of a Madin Darby Canine Kidney (MDCK) colony while cyclically stretched. Scale bar 50 μm. (c) Colony elongation ($Q_{xx}$) of control colonies along the elongation axis, control colonies in the laboratory framework (non-aligned, $\alpha = 0$) and colonies under cyclic uniaxial stretching in the laboratory framework (uniaxial stretching, $\alpha = 0$). Box Plot between 25th and 75th percentile, being the line in the box the median value, whiskers and outliers (dots) are obtained following Tukey's method, $N_{control} = 11$ independent experiments, n = 25 colonies and $N_{stretching} = 9$, n = 20 colonies. Mann-Whitney test control vs control aligned p=0.0003, control vs stretching p=0.0281 and control aligned vs stretching p=0.3319. (d) $Q_{xx}$ (left y axis) and $Q_{xy}$ (right y axis) during 120 min of colony expansion for control colonies and colonies under cyclic uniaxial stretching. Mean value ± standard error of the mean, $N_{control} = 8$, n > 14 colonies and $N_{stretching} = 9$, n = 20 colonies.

The online version of this article includes the following source data and figure supplement(s) for figure 2:

**Source data 1.** Raw data used for the panel (c) of *Figure 2* and $Q_{xx}(t)-Q_{xx}(0)$ and $Q_{xy}(t)-Q_{xy}(0)$ for the individual colonies used for panel (d) of *Figure 2*.
**Figure supplement 1.** Elongation as a function of frequency and amplitude of cyclic uniaxial stretching.
**Figure supplement 1—source data 1.** Values of $Q_{xx}(t)-Q_{xx}(0)$ and $Q_{xy}(t)-Q_{xy}(0)$ for each of the colonies included in *Figure 2—figure supplement 1*.

spontaneous elongation of colony when stretch was not applied, but elongation was oriented in the direction of externally applied uniaxial cyclic stretching (*Figure 2d*). Therefore, application of an external cyclic force can rectify symmetry breaking and set the direction of tissue elongation.

## Collective effects are essential for rectification

To get further insight into the collective nature of the rectification of tissue elongation, we probed the roles of adhesion between cells. First, we stretched single MDCK cells, individually plated. We observed that cells oriented perpendicularly to the externally applied uniaxial cyclic stretching (*Figure 3a and b*) as previously reported for fibroblasts (*Faust et al.,*

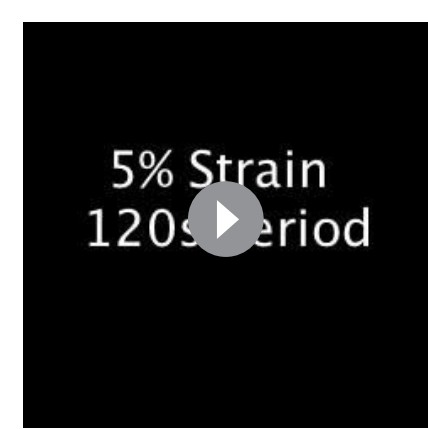

**Video 2.** Cyclic stretching of Madin Darby Canine Kidney (MDCK) colonies. Time-lapse of an MDCK colony under cyclic stretching (5% and 120 s). Time in hh:mm:ss. Scale bar 50 μm.
https://elifesciences.org/articles/57730#video2

*2011*). Then we blocked cell-cell junction in circular colonies prior stretching. Briefly, we incubated confluent colonies in medium containing 5 mM EDTA and 10 μg/ml anti-E-cadherin blocking antibody that targeted the extracellular domain of E-cadherin for 30 min (*Harris et al., 2014*). Then, medium was replaced by normal medium and the evolution of colonies with and without stretch was followed (*Figure 3c*). In the absence of externally applied uniaxial cyclic stretching, colonies treated with anti E-cadherin antibody expanded more than control colonies. Moreover, this expansion was still along one preferential direction (*Figure 3d*). Under cyclic uniaxial stretching, elongation was also non-isotropic and along the direction perpendicular to the cyclic stretching direction, in contrast to the parallel elongation observed when cell-cell contacts were intact (*Figure 2b–d*). This supports the collective nature of colony elongation. It is worth noting that cells inside the colony exhibited a decrease in their mean velocity (*Figure 3e*) and a large recruitment of myosin within cells similar to reinforcements (*Riveline et al., 2001*) in stretching conditions, as shown by the appearance of stress fibers (*Figure 3f*). However, this effect did not appear to affect the overall elongation rate. Altogether these data suggest that the asymmetric expansion of colonies in the direction imposed by cyclic uniaxial stretching is generally associated to a collective effect.

## Fingers and symmetry breaking

We next sought to identify the source of symmetry breaking in both conditions, with and without application of cyclic uniaxial stretching. It has previously been reported that in MDCK monolayers, cells can migrate tangentially to the monolayer boundary when confined (*Doxzen et al., 2013*), or perpendicular to the boundary in the form of multicellular groups or fingers during monolayer expansion (*Reffay et al., 2011*; *Reffay et al., 2014*). During spontaneous elongation of MDCK colonies in the absence of externally applied cyclic stretching (*Video 4*), we observed that boundary cells tend to move either perpendicularly or tangentially to the colony boundary (*Figure 4a*). In most of the experiments, an acto-myosin cable, similar to compartmentalization boundaries in vivo (*Monier et al., 2010*; *Calzolari et al., 2014*), was observed in the outer boundary of the colonies at stencil removal (see *Figure 4—figure supplement 1*). When this supra-cellular structure is intact, cells at the periphery are reported to undergo clockwise and counter clockwise rotations (*Doxzen et al., 2013*). In contrast, when a local interruption of this cable appeared, the cell at the vicinity could extend a lamellipodia and move away and radially from the center of the colony (*Figure 4—figure supplement 2a*). Apparently, a local defect in the cable could promote the local protrusion of a cell in the direction normal to the edge as shown in laser ablation experiments previously (*Reffay et al., 2014*). Several local defects could appear within the same colony, thus providing the opportunity for cells in the vicinity to protrude outwards. This cell has been termed leader cell (*Reffay et al., 2011*) and the collection of cells protruding from the circular colonies along this cell can be identified as the finger-like structures already reported for MDCK monolayers (*Reffay et al., 2011*; *Reffay et al., 2014*).

We performed cell tracking and observed that, on average, these protruding cells are faster than other boundary cells (*Figure 4a* and *Figure 4—figure supplement 2b and c*). They are characterized also by radial and directional migrations, in contrast to tangential motion observed in the other cells of the outer region of the colony (*Figure 4—figure supplement 2d and e*). In general, the motion of these so-called leader cells was directionally persistent and on average the shape of the whole colony followed the same overall direction (*Figure 4b and c*). To correlate colony elongation with leader cell orientation, we analyzed the evolution of a larger number of colonies for 2 hr after stencil removal (*Video 4*). We quantified the breaking of symmetry by fitting an ellipse to the shape of each colony. We then tracked the positions where finger-like structures were appearing, as well as the direction and distance performed by each of them. We could observe that the elongation direction of the whole colony correlated on average with the direction of the leader cell migration and associated finger (*Figure 4c*).

We then measured the position and displacement for each finger in control colonies and colonies under cyclic uniaxial stretching (*Figure 4d*). We observed that, when growing perpendicular to the direction of force application, finger cells performed shorter displacements than when growing parallel to it. In the absence of externally applied cyclic uniaxial stretching, fingers grew a similar amount as when growing parallel the direction of applied uniaxial cyclic stretching and no bias was observed vis-à-vis the nucleation position (*Figure 4d* and *Figure 4—figure supplement 2f*). To further explore this effect, we grew MDCK monolayers with straight boundaries either parallel or

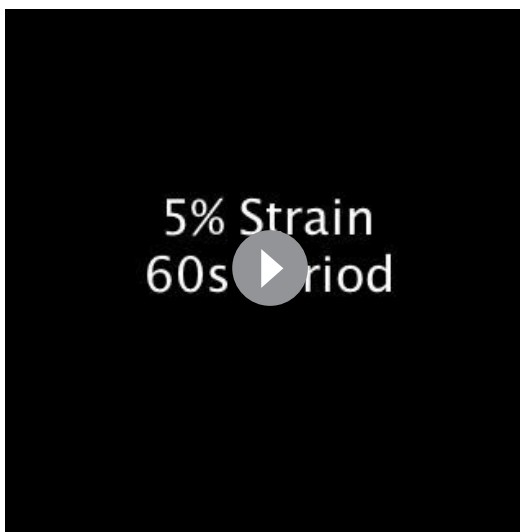

**Video 3.** Expansion of Madin Darby Canine Kidney (MDCK) colonies under cyclic stretching. Time-lapse of an MDCK colony under cyclic stretching (5% and 60 s) at the 0% strain position. Time in hh:mm. Scale bar 50 μm.

https://elifesciences.org/articles/57730#video3

perpendicular to the external force. Then, we let fingers appear and grow for 2 hr before applying cyclic uniaxial stretching (*Figure 4e*). When fingers were growing perpendicular to the stretching direction, they shrank upon application of cyclic uniaxial stretching; in contrast, fingers further elongated when parallel to the direction of uniaxial cyclic stretching. Altogether, this suggests that direction of finger-like structures correlates with elongation direction, and that external stretching affects the dynamics of finger growth.

## Collective effects and symmetry breaking

Finger growth correlates with colony elongation. However whether it is a cause or consequence of the symmetry breaking of the shape of the colony remains elusive. We therefore explored the possibility of inducing the growth of fingers and therefore set the direction of elongation of the colonies. Breakage of the acto-myosin cable by laser ablation induces the appearance of leader cells (*Reffay et al., 2014*). Hence we attempted to trigger the growth of fingers by locally injecting cytochalasin D using a micropipette. The transient injection of this actin polymerization inhibitor was followed by the disruption of the acto-myosin cable (*Video 5* and *Figure 5—figure supplement 1*). However, the cable reformed, and fingers did not appear. This result shows that breakage of the cable alone does not trigger the growth of fingers in our colonies, and suggests that other mechanisms may be involved.

We observed that when a finger moves outward from the colony, cells in the immediate vicinity elongate and seem to reorient their elongation axis toward the finger (*Figure 5a*). Recent studies have shown that the nematic field of cell elongation and its topological defects could be involved in the growth of bacterial colonies (*Doostmohammadi et al., 2016*) and in controlling dynamics, death and extrusion of epithelial cells (*Kawaguchi et al., 2017*; *Mueller et al., 2019*; *Saw et al., 2017*). We wondered if the spontaneous elongation of colonies would also be related to the average cell elongation. We followed the evolution of the cell elongation nematic field in different MDCK and MCF 10A colonies during expansion. We first obtained the spatio-temporal cell elongation nematic orientation field $\phi(x, y, t)$ (see Materials and methods) on the experimental time-lapse images (see *Figure 5b*, *Figure 5—figure supplement 2a-c*, *Video 6* and Appendix 1C). We could then obtain the angle $\theta_{\text{nematic}}$ of the average cell-shape nematic field at $t_0$ which we compared with final colony orientation $\theta_{\text{colony}}$ obtained using the ellipse fitting analysis (*Figure 5c*, *Figure 5—figure supplement 2d* and Appendix 1C). Strikingly, we observed a clear average cell elongation even at the time of stencil removal $t_0$, and the corresponding angle $\theta_{\text{nematic}}$ correlated with colony orientation when elongation direction was established for both MDCK and MCF 10A cell lines (*Figure 5d*, *Figure 5—figure supplement 2d and e*). The cell elongation orientation field $\phi(x, y, t)$ was not homogeneous at $t_0$ but exhibited a complex pattern with ±½ topological defects (*Figure 5b* and Appendix 1D). Interestingly, an expression that provides equilibrium orientation of liquid crystals with defects and having one constant Frank free energy,

$$\phi(x, y)_{\text{fit}} = \alpha + \sum_i k_i \tan^{-1}\left(\frac{y - y_i}{x - x_i}\right), \tag{1}$$

mimics the experimentally observed orientation field $\phi(x, y)$ very well with just one fitting parameter $\alpha$ (see Appendix 1D for details). Here $k_i = \pm\frac{1}{2}$ and $(x_i, y_i)$ are the strength and location, respectively, of the topological defects obtained from the experimental image. Thus, the defect position and

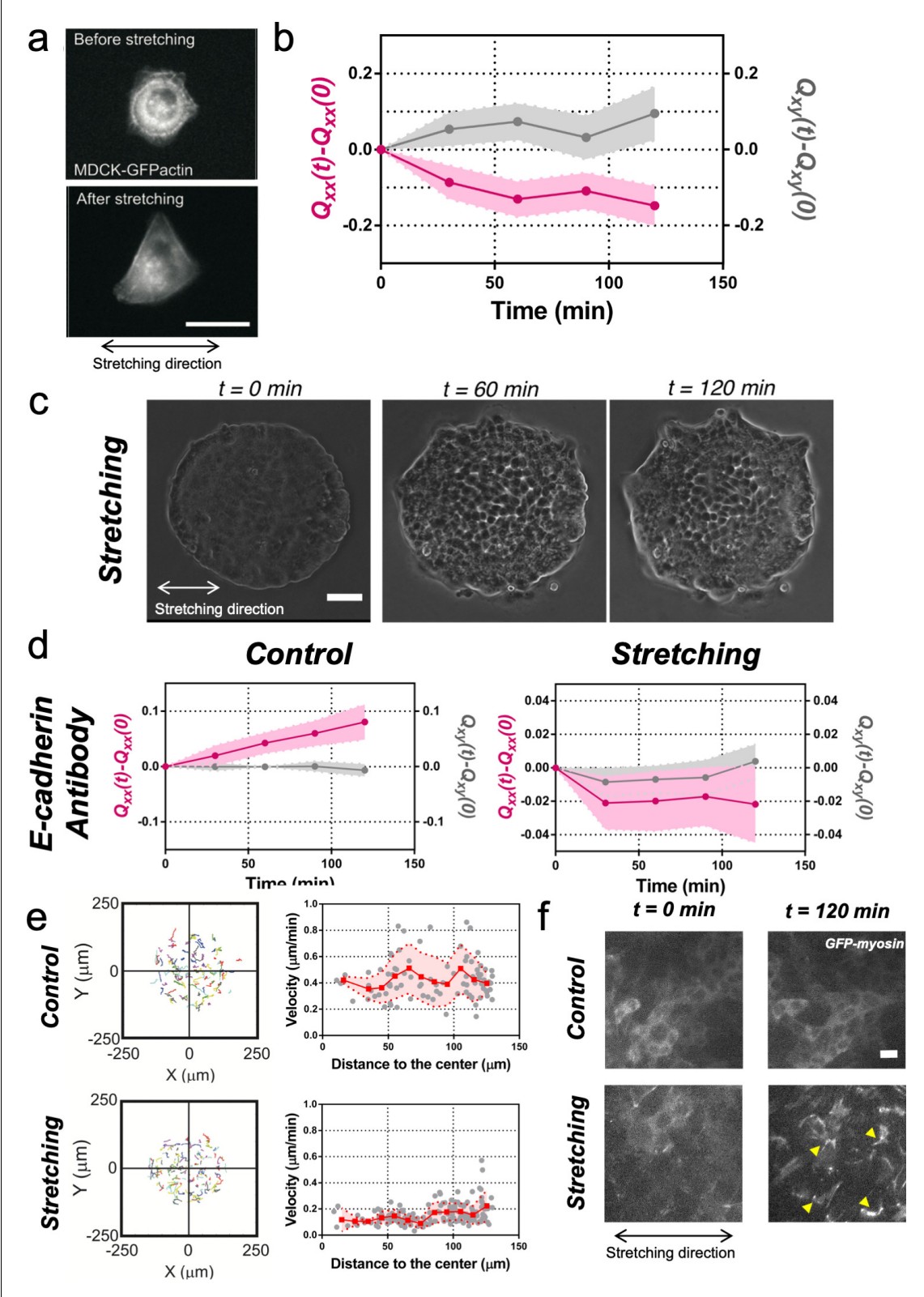

**Figure 3.** Collective effects are essential for rectification. (a) Image of a single MDCK-GFP-actin cell before and after being stretch for 2 hr (5% strain and 60 s period). Scale bar 20 µm. (b) $Q_{xx}$ (left y axis) and $Q_{xy}$ (right y axis) of single cells during 120 min of cyclic uniaxial stretching ($\alpha = 0$). Mean value ± standard error of the mean. N = 3, n = 31 cells. (c) Phase-contrast images of a Madin Darby Canine Kidney (MDCK) colony evolving for 120 min under cyclic mechanical stretching when E-cadherins are blocked by an E-cadherin antibody. Scale bar 50 µm. (d) Comparison of the cumulative $Q_{xx}$ (left y

*Figure 3 continued on next page*

*Figure 3 continued*

axis) and cumulative $Q_{xy}$ (right y axis) during 120 min of colony expansion when E-cadherin are blocked by an E-cadherin antibody in control and under cyclic uniaxial stretching ($\alpha$ = 0). Mean value ± standard error of the mean, $N_{control}$ = 3, n = 8 colonies and $N_{stretching}$ = 4, n = 15 colonies. (e) Trajectories of cells (left) and single-cell velocity as a function of its distance to the center of the colony (right) in control colonies (top) and under cyclic uniaxial stretching (bottom). $n_{control}$ = 90 cells from 8 colonies of $N_{control}$ = 4 independent experiments and $n_{stretching}$ = 154 cells from 13 colonies of $N_{stretching}$ = 4 independent experiments. Individual cells in gray, red square and line corresponds mean (binned by distance to the center), shadowed area corresponds to SD. (f) Myosin distribution inside MDCK-GFP-myosin colonies at 0 min and at 120 min after expansion in control and under uniaxial stretching. Note the myosin structures appearing in the stretching case (yellow arrows). Scale bar 10 µm.

The online version of this article includes the following source data for figure 3:

**Source data 1.** The $Q_{xx}(t)-Q_{xx}(0)$ and $Q_{xy}(t)-Q_{xy}(0)$ of individual cells used in panel (b) and individual colonies used in panel (d); we also provide datasets for the velocities of individual cells in panel (e).

strength can be used to provide an approximate readout for the orientation of the cell-shape nematic field (*Figure 5d*). Moreover, the location of finger nucleation seemed to be biased toward the position of topological defects. However, some defect locations were not stable in time and in some cases, the nematic field of cell shapes could only be interpreted in terms of virtual defects outside the colonies, thus suggesting that the mean nematic direction is a better readout for the cell-shape nematic field.

On the one hand, finger nucleation seemed to be correlated with colony elongation direction (*Figure 4c*). On the other hand, the orientation of tissue elongation correlates with orientation of average cell elongation at $t_0$ (*Figure 5* and *Figure 5—figure supplement 2*). This suggests that leader cells moving outward from the colony may not be the cause of symmetry breaking in colony shape, but rather follow from the initial cell shape elongation before stencil removal. Moreover, we found no correlation between breaks of the acto-myosin cable surrounding the colony and the mean nematic direction (*Figure 5—figure supplement 3*), which suggests that breaks are uniformly distributed along the colony border. We have also shown that breakage of acto-myosin cable after stencil removal, which is associated with leader cell formation (*Reffay et al., 2014*), did not necessarily induce the growth of fingers in our colonies. Altogether, our results could indicate that the orientation of the mean cell-shape nematic of the colony before stencil removal sets the direction of elongation by triggering the growth of fingers, which appear at the discontinuities of the outer acto-myosin cable located along the nematic orientation, while preventing finger growth at discontinuities located in other directions.

Finally, when looking at the evolution of the mean cell elongation nematic field of colonies under uniaxial cyclic stretching, we observed that it did not change over time (*Figure 5—figure supplement 4*). The initial mean direction of cell elongation, either parallel or perpendicular to the external stretching, was maintained throughout 2 hr of external stretching. This suggests that average cell elongation alone does not determine colony elongation direction when subjected to uniaxial cyclic stretching.

## Contributions to symmetry breaking

We next sought to evaluate quantitatively the contribution of cellular processes to elongation. We quantified the contributions of each cellular event using image segmentation, cell tracking, and neighbor triangulation (*Etournay et al., 2015*; *Etournay et al., 2016*) (see Appendix 1A, *Figure 6a and b*, *Figure 6—figure supplement 1* and *Figure 6—figure supplement 2*, and *Video 7*). This procedure decomposes overall tissue elongation, which is quantified in terms of cumulative total pure shear, into contributions

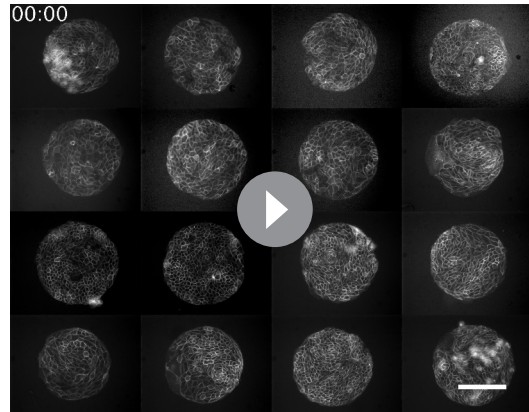

**Video 4.** Early symmetry breaking of circular colonies. Composite array time-lapse of MDCK-GFP-Ecadherin cells colonies during the first 2 hr of expansion. Time in hh:mm. Scale bar 150 µm.

https://elifesciences.org/articles/57730#video4

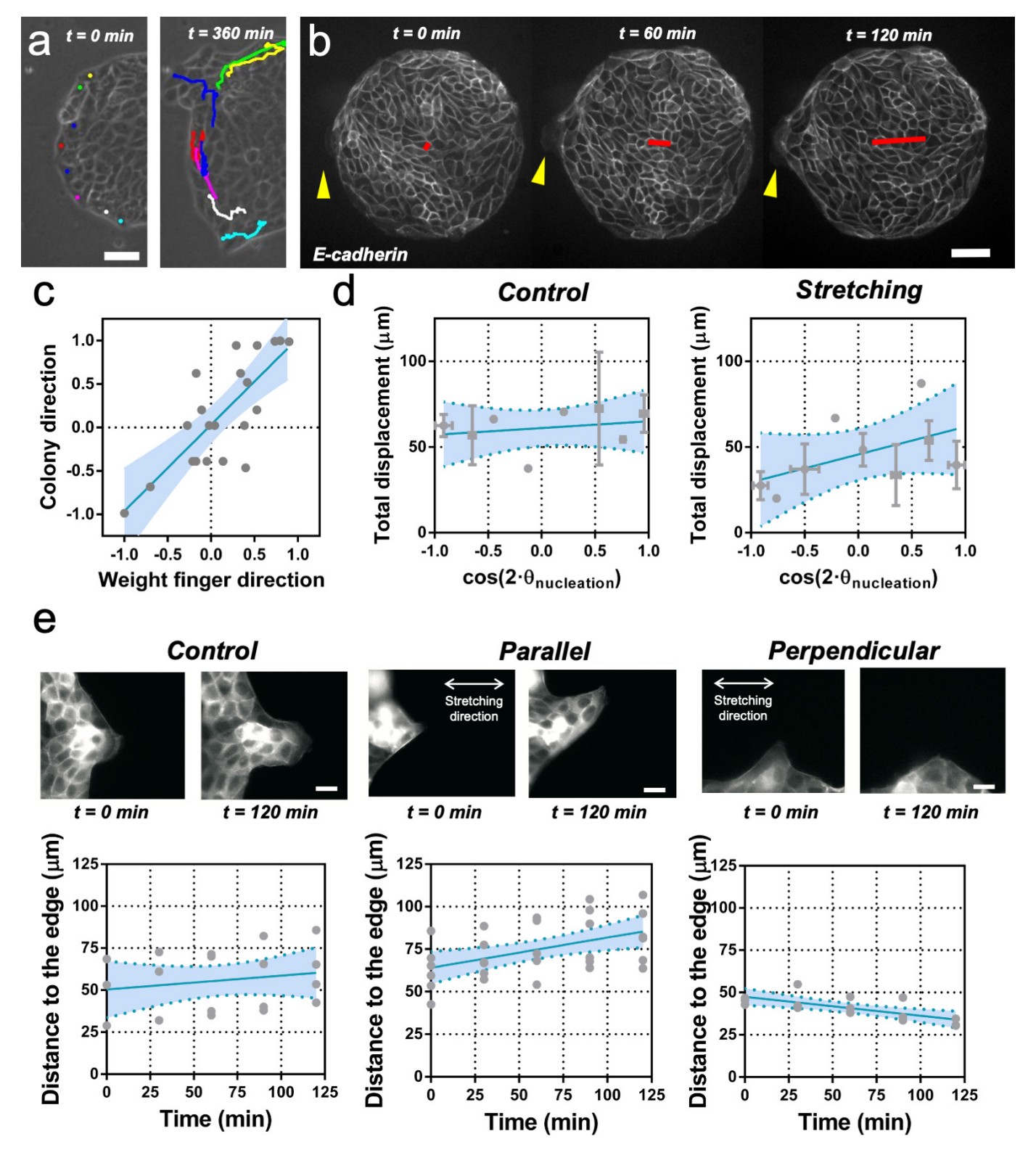

**Figure 4.** Fingers and symmetry breaking. (a) Trajectories of boundary cells during colony expansion. Two types of trajectories are observed: radial and tangential. Scale bar 50 µm. (b) Fluorescent images (GFP-E-cadherin) of a colony evolving for 120 min. Red line shows the orientation of the colony according to ellipse fitting (length scales with the change in $Q_{xx}$) and yellow arrow indicates a cell migrating radially. Scale bar 50 µm. (c) Direction of the colony elongation (quantified by the cosine of two times the angle of the main axis of the colony at t = 2 hr) as a function of the weight finger

*Figure 4 continued on next page*

*Figure 4 continued*

direction (quantified by the average of the cosine of two times the angle of each finger trajectory, for example the angle corresponding to the vector between the position of the finger at t = 0 hr and t = 2 hr, of each finger weight by the finger's displacement). N = 5, $n_{colonies}$ = 12 colonies and $n_{finger}$ = 21 fingers. Blue line corresponds to the linear fitting of the data points and the shadowed area corresponds to the 95% confidence interval. Pearson's correlation coefficient r = 0.7724, p=0.0001. (d) Total displacement of finger growth as a function of its initial position in the colony (angular coordinate from the center of the colony) for control colonies and colonies under uniaxial stretching. N = 5 independent experiments, $n_{colonies}$ = 11 colonies and $n_{finger}$ = 21 fingers (control) and N = 6, $n_{colonies}$ = 10 colonies and $n_{finger}$ = 28 fingers (stretching). Averaged fingers in gray (both position and distance, Mean ± SD), blue line corresponds to the linear fitting of the data points and the shadowed area corresponds to the 95% confidence interval. (e) Distance between the tip of the finger and the edge of the monolayer along time, for monolayers in control conditions, stretched parallel and perpendicular to the finger growth direction. $N_{control}$ = 3, $N_{parallel}$ = 4 and $N_{perpendicular}$ = 3 independent experiments and n = 4, 6, and 3 fingers, respectively. Individual fingers in gray, blue line corresponds to the linear fitting of the data points and the shadowed area corresponds to the 95% confidence interval.

The online version of this article includes the following source data and figure supplement(s) for figure 4:

**Source data 1.** Raw data corresponding to panels (c), (d) and (e).
**Figure supplement 1.** A supra-cellular acto-myosin cable sets the boundary of the colony.
**Figure supplement 2.** Leader cell dynamics.
**Figure supplement 2—source data 1.** Datasets for the individual velocities, angles and directionality index plotted in panels (c–f).

from cell elongation and topological changes. Five main events contribute to total shear: cell elongation change, cell division, cell extrusion, T1 transition, and correlation effects (*Etournay et al., 2015*; *Etournay et al., 2016*). At the colony scale, shear decomposition plots (*Figure 6c* and *Figure 6—figure supplement 3*) revealed that the total pure shear gives values consistent with elongation estimates from ellipse fitting (*Figure 6d*). Note that various contributions to shear decomposition exhibit significant variability between experiments (*Figure 6—figure supplement 3*). However, we found that after the first 2 hr, the contribution of cell elongation is generally comparable to the total pure shear, with a smaller contribution from other sources (*Figure 6d*). When looking at the shear decompositions of colonies under cyclic uniaxial stretching (*Figure 6e* and *Figure 6—figure supplement 3*), the cumulative shear values were also similar to the ones obtained by ellipse fitting (*Figure 6f*). Interestingly, we found however that in that case, shear created by T1 transitions is the main contributor for the total pure shear, while cell elongation does not contribute to tissue elongation (*Figure 6f* and *Figure 6—figure supplement 3*). This indicates that applying oscillatory forces to the tissue changes fundamentally the main mode of tissue elongation by favoring topological rearrangements of the cell network.

## Vertex model recapitulates symmetry breaking and shear decomposition

We then asked whether a model could reproduce experimental observations of shear decomposition and, in particular, in which conditions tissue elongation would arise from cell elongation or from topological transitions. We developed a vertex model which takes into account mechanical anisotropies such as active stresses and polarized cell bond tension (see *Figure 7a* and Appendix 2). We generated a confluent colony of circularly confined cells, in which a unit director **p** that represented the cell polarity was assigned to each cell. Based on orientation of the director, each cell generated an extensile active stress $\sigma_a$ and bias $\lambda$ in the base value of its edge contractility to promote cell elongation and active T1 transitions. We assumed that the experimentally measured cell elongation nematic **q** is a readout of the underlying cell polarity **p** (*Figure 7b*). Hence, the initial spatial distribution of **p** was based on the commonly observed pattern of **q**

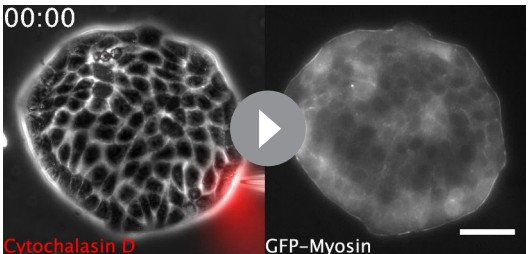

**Video 5.** Disruption of the acto-myosin cable. Movie showing the disruption of the acto-myosin cable. The colony and the micropipette used are shown at the left and the myosin signal is shown at the right. Cytochalasin D was mixed with Cy5 to allow its visualization. Time in hh:mm. Scale bar 50 μm (right).
https://elifesciences.org/articles/57730#video5

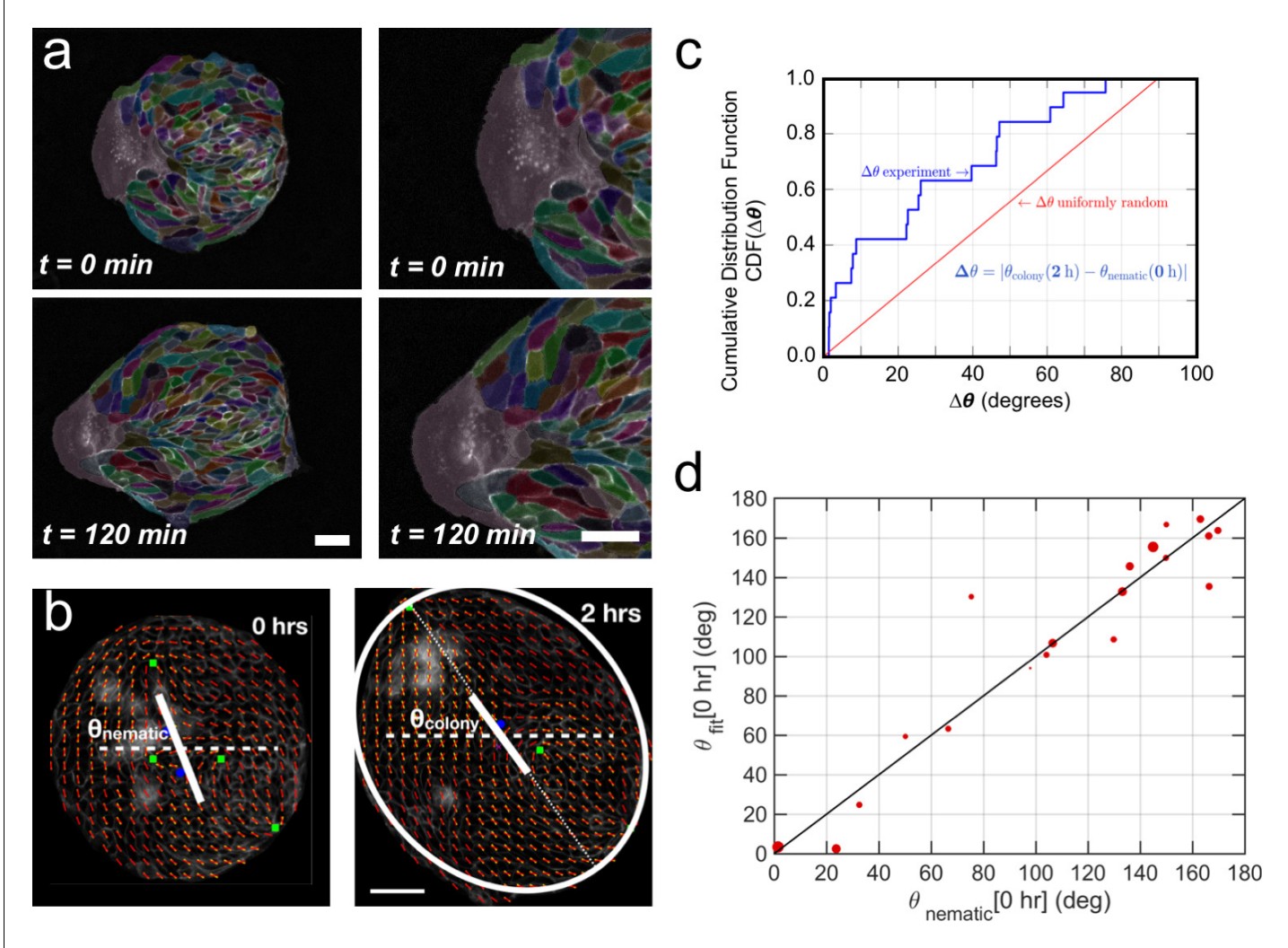

**Figure 5.** Collective effects and symmetry breaking. (a) A leader cell at the boundary of the colony pulls the colony outwards while inner cells deform and elongate. Scale bar 50 µm. (b) Cell shape is quantified using a nematic field (red segments). First, the mean cell shape nematic is quantified at the moment of stencil removal (0 hr) and the orientation $\theta_{nematic}$ of its mean for the entire colony is obtained. Then, the overall shape of the colony after 2 hr is obtained by fitting an ellipse, whose major axis makes an angle $\theta_{colony}$. The yellow directors correspond to fits for the cell shape nematic field obtained with respect to the +1/2 and −1/2 topological defects of the experimentally obtained (red) nematic field (also see Appendix 1D for details.) Scale bar 50 µm. (c) The cumulative distribution function (CDF) for the difference $\Delta\theta$ between $\theta_{nematic}$ (0 hr) and $\theta_{colony}$ (2 hr) is obtained from n = 19 colonies of N = 5 independent experiments. Red line corresponds to the CDF of a random distribution of the difference $\Delta\theta$. This plot shows a strong correlation between the cell shape nematic and the overall shape symmetry breaking (also see *Figure 5—figure supplement 2d*, and Appendix 1E). (d) The experimentally measured angle of mean nematic orientation $\theta_{nematic}$ obtained for 19 colonies at t = 0 hr is compared with its counterpart $\theta_{fit}$ obtained by fitting the experimental data with *Equation 1* of the main paper with respect to the orientation parameter α (see Appendix 1D). The size of the red circles in (b) is proportional to the magnitude of anisotropy of the colony shape after 2 hr. n = 19 colonies of N = 5 independent experiments for MDCK.

The online version of this article includes the following source data and figure supplement(s) for figure 5:

**Source data 1.** Raw data corresponding to panels (c) and (d).

**Figure supplement 1.** Supra-cellular acto-myosin cable disruption.

**Figure supplement 2.** Cell shape nematic field.

**Figure supplement 2—source data 1.** Values of the probability to obtain the experimental cumulative distribution function of *Figure 5c* and we list the values for the x-axis, the y-axis and the weight associated to each point of *Figure 5—figure supplement 2e*.

**Figure supplement 3.** Mean nematic direction and defects in the acto-myosin cable.

**Figure supplement 3—source data 1.** Values of the experimental data points used to obtained the cumulative distribution function showed in *Figure 5d*.

**Figure supplement 4.** Uniaxial cyclic stretching does not modify the mean nematic direction.

*Figure 5 continued on next page*

*Figure 5 continued*

**Figure supplement 4—source data 1.** Individual values represented in *Figure 5b*.
**Figure supplement 5.** Obtaining topological charge for the orientation field *q* on a rectangular grid.

(*Figure 7c*). To evolve **p** with time, we imposed that **p** of the exterior cells tended to be parallel to the boundary, whereas the inner cells tended to align their **p** with those of their neighbors (Appendix 2D).

Upon removal of confinement, we found that the simulated colony spontaneously elongates, in a direction set by the orientation of the mean cell elongation nematic field, along with the formation of finger-like structures near the +½ defect, as observed experimentally (see *Video 8*). Our simulations therefore reproduce experimental observations indicating that colony deformation can be understood without forces exerted by a leader cell at the colony boundary (*Figure 7c*, *Video 8*, and Appendix 2). To test whether we could also recapitulate different contributions of the total pure shear inside the tissue, we performed a shear decomposition analysis of in silico movies. We found a qualitatively similar cumulative shear-strain decomposition as was observed in experiments (*Figure 7d and e*), where the main contribution came from cell elongation. Moreover, by changing the relative contributions of the cellular active stress magnitude ($\sigma_a$) and the edge tension bias ($\lambda$), we could modulate the relative contributions from cell elongations and T1 transitions to the total pure shear (*Figure 7—figure supplement 1*) as was also observed in experiments with colonies in the absence or presence of cyclic stretching (*Figure 6—figure supplement 3*). When $\sigma_a$ was dominant, the colony elongation was primarily due to cell elongation, whereas when $\lambda$ was the stronger term, T1 transitions were the main cause of colony elongation. These results reveal possible cellular mechanisms that can govern the process of tissue deformation and influence whether cell elongation, or cellular rearrangements, dominate tissue elongation.

Our vertex model assumed that the cell elongation was the main readout for cell polarity, and it did not explicitly account for the effect of substrate stretching. To incorporate uniaxial cyclic stretching, we further developed the model. Our results show that initial cell shape elongation does not have a preferential direction (*Figure 5—figure supplement 4*), but colony elongation under uniaxial cyclic stretching is along *x* axis, the direction of stretching, and mainly achieved through T1 transitions (*Figure 2c* and *Figure 6f*). Also, we report that the elongation happens along *y* axis, perpendicular to the direction of stretching, in single cells and cell colonies with lowered Ecadherin levels (*Figure 3a–d*). These two experimental observations can be implemented in the model. First, we introduced an additional term $\boldsymbol{m}_{stretch}$ that oriented cell polarization **p** along *x* axis, for any given initial condition, upon confinement removal at $t_0$ – this term is inactive in the absence of uniaxial stretching (Appendix 2D). Then, by using cell active stress $\sigma_a > 0$, we mimicked the tendency of single cells to elongate perpendicular to the orientation of polarity, i.e., perpendicular to the uniaxial stretching. In contrast, the bias in the edge tension $\lambda > 0$ induced T1 transitions along the polarity of the cell, i.e., parallel to the uniaxial stretching. Thus, in the presence of uniaxial stretching, $\boldsymbol{m}_{stretch}$ oriented cell polarities along *x*, while the relative magnitudes of single cell active stress $\sigma_a$ and edge-contractility bias $\lambda$ dictated the orientation of the colony elongation. When keeping a lower magnitude of $\sigma_a$ and a higher value of $\lambda$, colonies elongated along *x* through T1 transitions (*Figure 7f* and *Video 9*), mimicking colony elongation under uniaxial cyclic stretching (*Figure 2*

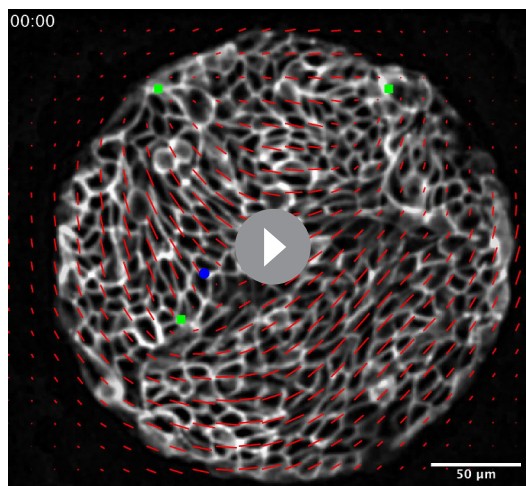

**Video 6.** Nematic field alignment precedes colony elongation. Movie showing cell-shape nematics and topological defects of an elongating colony. Time in hh:mm. Scale bar 50 µm.
https://elifesciences.org/articles/57730#video6

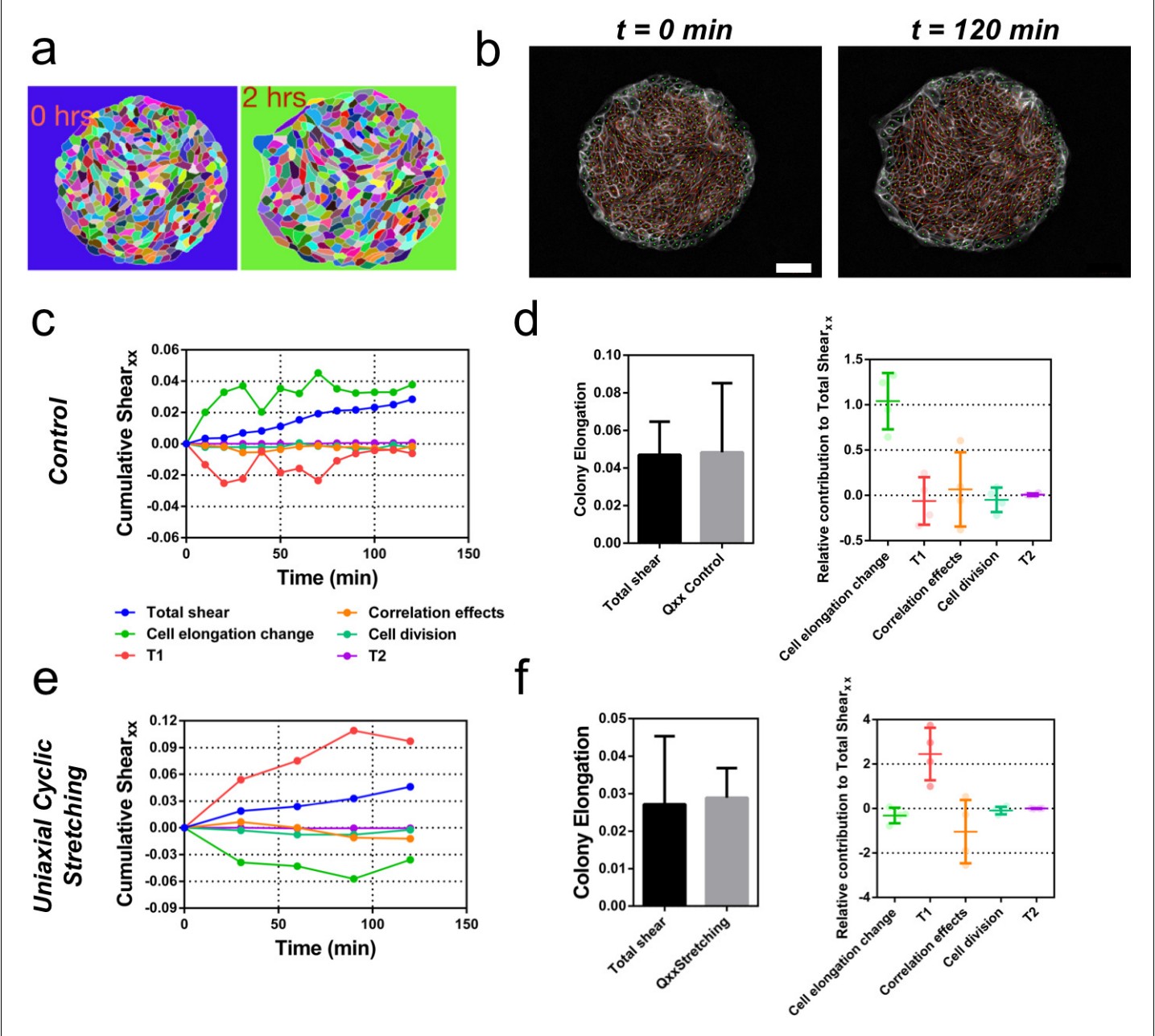

**Figure 6.** Contributions to symmetry breaking. (**a**) Snapshots of two colonies from control condition segmented and tracked for cells using Tissue-Analyzer (TA) for a duration of $t$ = 120 min starting from removal of stencil at $t$ = 0. Scale bar 50 µm. (**b**) The segmented and tracked images are triangulated in Tissue-Miner (TM). Scale bar 50 µm. The green dots represent the centers of the segmented cells. (**c**) The dynamics of triangulation is analyzed in TM to provide the overall $xx$ component of cumulative pure shear strain in a sample colony as a function of time (total shear). (**d**) Comparison between the mean total pure shear obtained from TM and the overall colony pure shear obtained from ellipse fitting (left). Total shear corresponds to $n_{colonies}$ = 4 colonies from N = 2 independent experiments and $Q_{xx}$ control was obtained from $n_{colonies}$ = 25 colonies of N = 11 independent experiments. Relative contribution of the different processes to the total pure shear (right). Total shear and contributions were obtained from $n_{colonies}$ = 4 colonies from N = 2 independent experiments. (**e**) Cumulative pure shear decomposition for stretched colony. (**f**) Comparison between the mean total pure shear obtained from TM and the overall colony pure shear obtained from ellipse fitting (left) and relative contribution of the different processes to the total pure shear (stretching case). Total shear corresponds to $n_{colonies}$ = 4 colonies from N = 4 independent experiments and $Q_{xx}$ stretching was obtained from $n_{colonies}$ = 20 colonies of N = 9 independent experiments. Relative contribution of the different processes to the total pure shear (right). Total shear and contributions were obtained from $n_{colonies}$ = 4 colonies from N = 4 independent experiments.

The online version of this article includes the following source data and figure supplement(s) for figure 6:

**Figure supplement 1.** Steps in image analysis.

*Figure 6 continued on next page*

and *Figure 6f*). On the contrary, by increasing σ$_a$ and lowering λ (Ecadherin deficient colonies), colonies elongated perpendicular to $x$ (*Figure 7g* and *Video 9*), mimicking colonies under uniaxial cyclic stretching treated with E-cadherin antibody (*Figure 3c–d*). Therefore, we propose that a competition between the strength of active T1 transitions parallel to the external stretching and active cell stress perpendicular to the external stretching dictate overall colony elongation under uniaxial cyclic stretching. When cell-cell junctions are intact, colony elongation is along the direction of stretching and through T1 transitions (*Figure 2c* and *Figure 6f*), suggesting that the tendency of single cells to orient along $y$ (*Figure 3a–b*) is partially screened by cell-cell junctions via T1 transitions. When cell-cell junctions are weakened, active cell stress dominates, and colonies, which could be thought of to be closer to a collection of single cells, elongate perpendicular to the uniaxial stretching direction (*Figure 3c and d*).

## Stretching-dependent elongation is mediated by ROCK

We showed that upon stretching, cells reduced their speed and myosin structures appeared (*Figure 3e and f*). These type of cellular responses to external stretching involve the Rho-associated protein kinase (ROCK) (*Hart, 2020*), which is also involved in cell-cell contacts integrity (*Ewald et al., 2012*; *Nishimura and Takeichi, 2008*). We treated MDCK colonies with a ROCK inhibitor (Y-27632 50μM) and followed their behavior for 2 hr, both in the absence and the presence of cyclic uniaxial stretching (*Figure 8—figure supplement 1*). When cyclic stretching was applied, elongation along the direction of the uniaxial cyclic stretching was absent ($Q_{xx} \approx Q_{xy}$) (*Figure 8—figure supplement 1a and b*). However, colonies still elongated anisotropically, similar to colonies in the absence of application of uniaxial cyclic stretching (*Figure 8—figure supplement 1c*). According to our model, when edge tension bias is sufficiently large, the dominant mechanism for tissue elongation switches from single-cell elongations to T1 transitions (*Figure 7—figure supplement 1*). We observed that uniaxial cyclic stretching triggers this type of elongation (*Figure 6f*). Therefore, if the effect that application of uniaxial cyclic stretching has at the cellular level was reduced, colonies subjected to cyclic uniaxial stretching should preferentially elongate as if cellular active stress becomes dominant, that is through single-cell elongation. Strikingly, shear decomposition analysis shows that the elongation mechanism shifts from T1 transition-dominant to cell elongation-dominant when colonies under uniaxial cyclic stretch have ROCK inhibited (*Figure 8a-b*). In summary, colonies under cyclic uniaxial stretching elongate through T1 transitions rather than through cell elongation (red dot in *Figure 8c*) similar to what our model predicts for colonies with increased biased edge tension. In contrast, colonies elongate spontaneously largely through single-cell elongation, which the model predicts when the effect of the cellular active stress is more dominant (green dot in *Figure 8c*). Strikingly, the application of a ROCK inhibitor leads to single-cell elongation dominating over T1 (orange dot in *Figure 8c*), effectively suppressing the effect of uniaxial cyclic stretching on the mode of colony deformation (*Figure 8—figure supplement 1*).

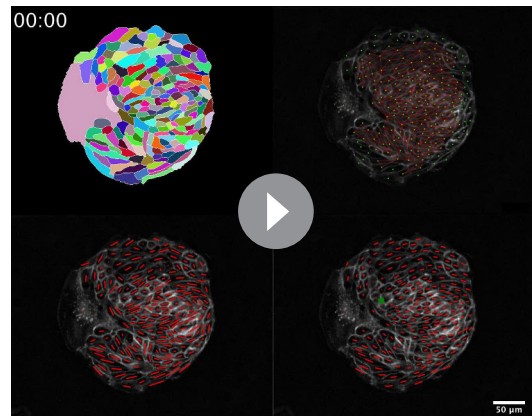

**Video 7.** Contributions to colony elongation. Movie showing cell tracking, neighbor triangulation, cell-shape nematics, and topological defects of an elongating colony. Time in hh:mm. Scale bar 50 μm.

https://elifesciences.org/articles/57730#video7

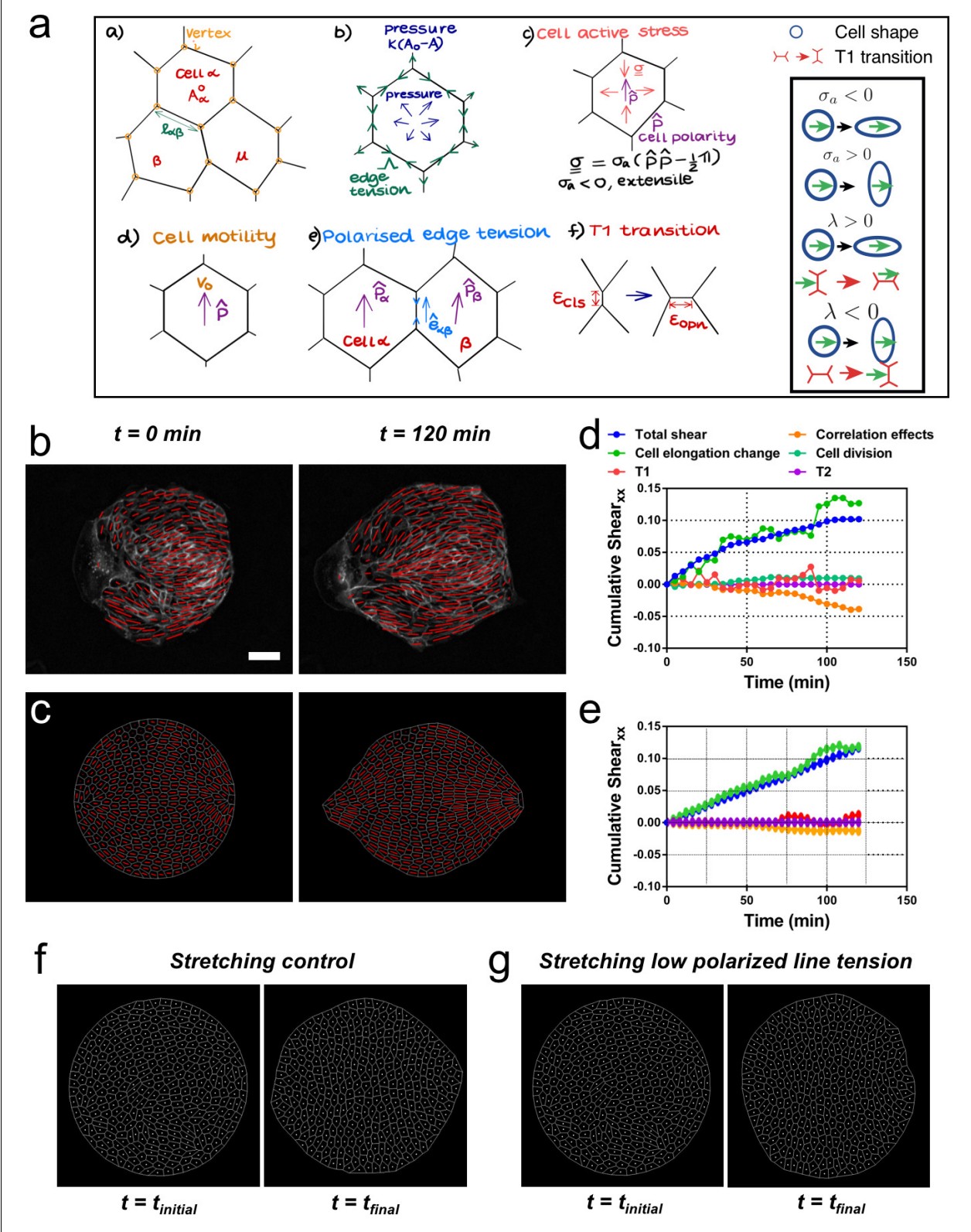

**Figure 7.** Vertex model recapitulates symmetry breaking and shear decomposition. (a) Schematic of vertex model depicting the arrangement of cells, forcing, and topological transitions in a tissue. 2-D monolayer of epithelial cells is represented by polygons, generally sharing a common edge and two vertices between cells. For any cell α shown in the figure, $A_\alpha$ is the area, $A_\alpha^0$ is the preferred area and $l_{\alpha\beta}$ is the length of the edge shared between cells α and β. The forces on any vertex $i$ in the basic vertex model are from pressure due to deviation in the cell area from its preferred value and the tensile

*Figure 7 continued on next page*

*Figure 7 continued*

force arising from edge or cortical contractility Λ. In our model, each cell also has a polarity *p* associated with it through which active forces can act on the cell vertices due to anisotropic cell active stress (extensile in our case), cell motility $v_0$ and polarized or biased edge tension that depends on the orientation of the edge $e_{\alpha\beta}$ with respect to the polarities of the adjoining cells. When the edge connecting two cells becomes smaller than a critical value $\epsilon_{cls}$, the cells are made to modify their neighbors by forming a new edge of length $\epsilon_{opn} > \epsilon_{cls}$ as shown. Scheme depicting the different possibilities from the model parameters. (b) Experimental coarse-grained cell shape nematic at *t=0* and *t=120* min. Scale bar 50 µm. (c) A vertex model with internal activity arising from extensile active cell stress ($\sigma_a'$ = -5) and biased edge tension for the cell-cell junctions ($\lambda'$ = 20). Prime symbol ′ refers to non-dimensional values (see Appendix 2E). (d) Overall *xx* component of cumulative pure shear strain in the sample colony shown in (b) as a function of time (total shear). (e) Shear decomposition of the *in silico* experiment which is similar to its experimental counterpart in (d). (f) A vertex model with an additional aligning term that defines the direction of the uniaxial stretching through cell polarity. When cell-cell junctions are intact, biased edge tension ($\lambda'$ = 50) dominates over active cell stress ($\sigma_a'$ = 2) and the colony elongates along the (horizontal) direction of stretch collectively through T1 transitions. (g) When the effect of edge tensions is lowered ($\lambda'$ = 25) and active cell stress is increased ($\sigma_a'$ = 4), the colony elongates perpendicularly to the direction of stretch.

The online version of this article includes the following source data and figure supplement(s) for figure 7:

**Figure supplement 1.** Cumulative pure shear decomposition patterns for the simulated colonies.
**Figure supplement 1—source data 1.** Values of each of the different contributions to cumulative shear along time for the different simulations.

## Discussion

Tissue spreading is key during embryonic development (*Bénazéraf et al., 2010*; *Campinho et al., 2013*; *Etournay et al., 2015*). Epithelial cells migrate in a collective and cohesive manner. In many cases symmetry is broken leading to shape transformation. The resulting tissue kinematics and the underlying mechanisms for this symmetry breaking has been studied using original approaches, and is understood with a variety of cell-based and continuum models in vivo (*Blanchard et al., 2009*; *Ciarletta et al., 2009*; *Aigouy et al., 2010*; *Etournay et al., 2015*; *Guirao et al., 2015*; *Alt et al., 2017*). On the in vitro situation too, combination of theory and experiments have been applied on similar phenomena (*Mark et al., 2010*; *Tarle et al., 2015*; *Kawaguchi et al., 2017*; *Saw et al., 2017*).

In the present work, we sought to integrate knowledge from in vitro and in vivo studies to test new ideas for breaking symmetry through collective effects. First we showed that initially circular colonies of three different epithelial cell lines spontaneously expanded in a non-isotropic manner and the elongation observed was similar in magnitude to *Drosophila* wing blade elongation in vivo (*Etournay et al., 2015*). However, the undeformed circular colonies have a non-zero average cell elongation even before spreading starts, which determines orientation of the final colony shape. Our analysis also showed that the cell orientation patterns are not homogeneous but spatially organized and directed by the presence of ±½ topological defects. It was already shown that topological defects in the cell elongation nematic field have a key role on epithelial dynamics (*Kawaguchi et al., 2017*) and on cell death and extrusion (*Saw et al., 2017*). Our results reinforce the idea that cell elongation nematic field, which could only arise from collective interaction between cells, can have an impact on epithelial morphogenesis.

Since developing tissues are regularly subjected to internal oscillations (*Solon et al., 2009*; *He et al., 2010*; *Rauzi et al., 2010*) and external pulsatory forces (*Zhang et al., 2011*; *Hayes and Solon, 2017*) in a number of living organisms, we explored the effect of external forces in the in vitro circular colonies. We observed that the direction of elongation could be rectified by imposing an external uniaxial cyclic force. This particular behavior is of great interest from an in vivo point of view: cyclic contraction could direct elongation in specific directions to trigger the next steps in morphogenesis. The generic localizations of muscles connected to epithelial layer (*Zhang and Labouesse, 2012*) could have this essential function for tissues which would otherwise elongate in any direction like in our assay.

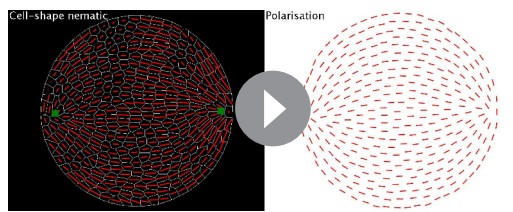

**Video 8.** In silico recreation of a colony elongation. Vertex model simulation of a colony elongation. Cell-shape nematics, topological defects, and cell polarization are followed over time.
https://elifesciences.org/articles/57730#video8

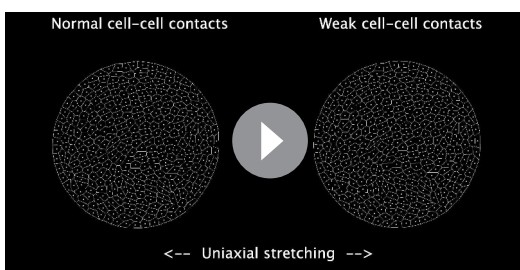

**Video 9.** In silico recreation of single versus collective effect of stretching. Competition between active cell stress $\boldsymbol{\sigma}_a$ and biased junction tension $\lambda$ governs direction of colony elongation. (Left) When cell-cell junctions are normal, we propose that $\lambda'=50$ dominates over $\boldsymbol{\sigma}_a' = -2$ and the colony elongates along the direction of stretch (horizontal) collectively through T1 transitions (collective stretching). (Right) When E-cadherin levels are low (blocked by anti-E-cadherin antibody), the effect of edge tensions $\lambda'=25$ is lowered as compared and that of active cell stress $\boldsymbol{\sigma}_a' = -4$ is increased, thus leading to elongation of the colony perpendicular to the direction of stretch (vertical) through individual cell elongation (single stretching). The superscript ' indicates non-dimensionalized parameter (see Appendix 2E).
https://elifesciences.org/articles/57730#video9

Finally, direction of this tissue rectification is along the external force, but perpendicular to the reorientation of single cell under external uniaxial stretching, and this further supports the collective nature of the phenomenon.

We also systematically quantified the shear deformation kinematics of the colony and demonstrated that some of the colonies exhibit shear decomposition patterns similar to those observed during *Drosophila* pupal wing elongation (*Etournay et al., 2015*). Moreover, while the colony deformation for the control colonies was dominated by cell elongation, T1 transitions were the main drivers of the colony shape anisotropy under cyclic uniaxial stretching thus indicating different mechanisms at work. Thus, the current work thus makes a direct comparison and contrast tissue kinematics between in vivo and in vitro cases.

Finally, we developed a vertex model that takes into account mechanical anisotropies. Cell anisotropic active stress arising in the cell core, cortical contractility at the cell-cell junctions and cell motility are three of the important forces involved in morphogenesis of epithelial monolayers. To our knowledge, this is the first attempt to systematically show how each of these activities acts on tissue kinematics. Specifically, we showed that cell anisotropic active stress results mainly in cell elongation, whereas anisotropies in cortical contractility primarily effects cell intercalations or T1 transitions. Including cell motilities appears to enhance tissue shear generated by the other two modes of internal forcing. We perturbed active stress and line edge tension by inhibiting ROCK, which has been reported to be involved in cell-cell contact integrity in vivo (*Nishimura and Takeichi, 2008*; *Ewald et al., 2012*), and recently, in cell responses to stretching in vitro (*Hart, 2020*). This led to experimentally blocking the ability of colonies to respond to the externally applied uniaxial cyclic stretching. By doing so, colonies which primarily elongate through cell intercalations, shifted to a single cell elongation driven mechanism.

From our simulations, we could demonstrate that symmetry breaking and finger formation in colonies and the corresponding tissue kinematics observed in our experiments could be brought about by collective active behavior of the colony cells and does not require special action of leader cells (*Theveneau and Linker, 2017*). This result echoes experiments in which the emergence of the leader cells and the fingering behavior at the border were suggested to arise due to the internal stress pattern in the tissue (*Vishwakarma et al., 2018*). On the other hand, there are many excellent models in which the boundary cells are ascribed special motility properties that could replicate the experimental results on border fingering (*Mark et al., 2010*; *Tarle et al., 2015*). Thus, although leader cells at the boundary may play a role in the border fingering, our experimental findings and simulations clearly indicate that the cell-level internal activities and cell-cell interactions are sufficient to cause symmetry breaking in the colony shape and its overall kinematics via the collective cell-shape nematic field.

## Conclusion

Our results show that cell elongation nematic field can have an impact on epithelia morphogenesis. It was already reported that topological defects in the cell elongation nematic field have a key role on epithelial dynamics (*Kawaguchi et al., 2017*) and on cell death and extrusion (*Saw et al., 2017*). Now, we showed that circular epithelial colonies when in confinement build up a mean nematic orientation. This symmetry breaking results from the inner activity of cells, and sets the direction for

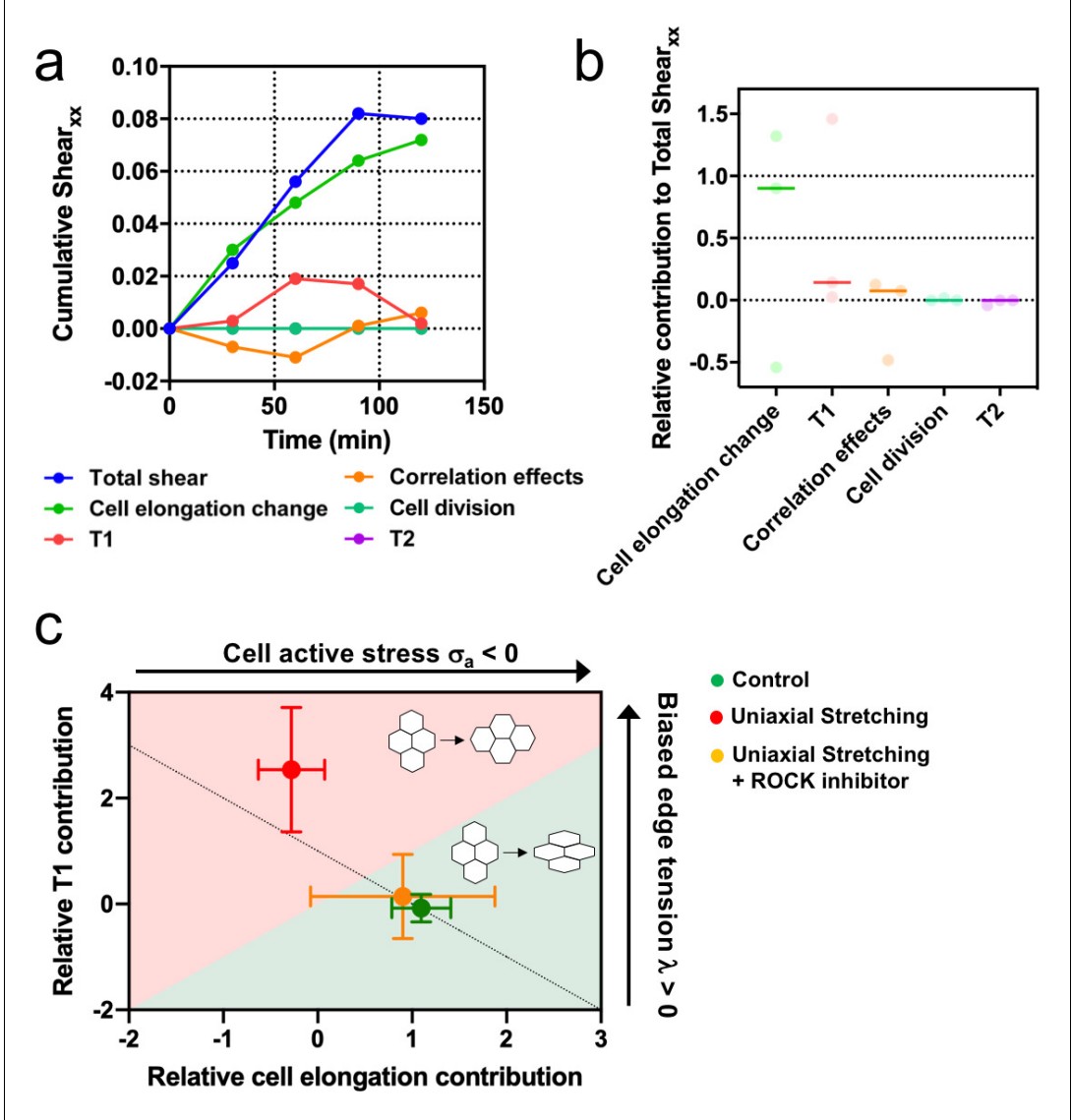

**Figure 8.** Stretching-dependent elongation is mediated by ROCK. (**a**) Cumulative pure shear decomposition for stretched colony in the presence of ROCK inhibitor Y-27632 50 μM. (**b**) Relative contribution of the different processes to the total pure shear. Total shear and contributions were obtained from $n_{colonies} = 3$ colonies from $N = 3$ independent experiments. Individual experiments and median are plotted. (**c**) Contribution of single-cell elongation to total elongation versus contribution of T1 transitions to total elongation. In the green area, single-cell elongation dominates over T1, whereas red area corresponds to T1 transitions dominating over cell elongation. Along the dashed line, contribution of correlation effects and oriented cell division to total shear is close to zero. Above this line this contribution is negative and below the line is positive. According to our vertex model, by changing the strength of cellular active stress and biased edge tension, colonies change the relative contribution of each mechanism of elongation. Experiments show that epithelial colonies spontaneously elongate through single-cell elongation (green point, $N = 2$, $n_{colonies} = 4$), whereas colonies under uniaxial stretching elongate through T1 (red dot, $N = 4$, $n_{colonies} = 4$). When ROCK activity is inhibited, colonies under uniaxial stretching elongate through single-cell elongation (orange dot, $N = 3$, $n_{colonies} = 3$). Points represent median ± SD.

The online version of this article includes the following source data and figure supplement(s) for figure 8:

**Source data 1.** Values of each of the different contributions to cumulative shear as time progresses for the three colonies analyzed.

**Figure supplement 1.** Stretching-dependent elongation is mediated by ROCK.

**Figure supplement 1—source data 1.** Values of $Q_{xx}(t)-Q_{xx}(0)$ and $Q_{xy}(t)-Q_{xy}(0)$ for each of the colonies included in *Figure 8—figure supplement 1*.

colony elongation. Epithelia changes in shape could be revisited in vivo with this new framework, leading to potential generic rule of morphogenesis in developmental biology.

## Materials and methods

### Cell culture

MDCK cells (GFP-E-cadherin strain [*Adams et al., 1998*], GFP-Actin strain, mCherry-actin / GFP-myosin strain [*Klingner et al., 2014*]) were cultured in Dulbecco's modified Eagle medium (D-MEM) 1 g/l glucose (Invitrogen), supplemented with 10% fetal bovine serum (FBS) (Invitrogen) and 1% penicillin-streptomycin (and the corresponding resistance for each strain). Cells were passaged every 2–3 days using standard procedures. Caco-2 cells (ATCC) were cultured in minimum essential media (MEM) supplemented with Earle's salts (Life Technologies), 20% fetal calf serum (FCS) (Invitrogen), 0.1 mM non-essential amino acids, 1 mM sodium pyruvate, and 40 µg/ml Gentamicin at 37°C and 5% $CO_2$. Culture was passaged every 3–4 days. MCF 10A cells (ATCC) were cultured in Dulbecco's modified Eagle medium (D-MEM) 1 g/l glucose (Invitrogen), supplemented with 10% horse serum (Invitrogen), 5 µg/ml insulin, 40 µg/ml Gentamicin, 2 mM L-Glutamine, 0.5 µg/ml Hydrocortisone, and 2 ng/ml human epidermal growth factor (hEGF). Cells were passaged every 2–3 days. Cells tested negative for mycoplasma.

### Fabrication of PDMS membranes and stencils

Poly(dimethylsiloxane) (PDMS) (Sylgard) was prepared by mixing the pre-polymer and the cross-linker at a 10:1 ratio. To prepare stretchable membranes, uncured PDMS was first centrifuged (3000 rpm for 5 min) to remove air bubbles. Afterwards, the PDMS was spin-coated on a flat polystyrene (PS) surface (500 rpm for 1 min) and cured at 65°C overnight. PDMS stencils were prepared as described previously (*Ostuni et al., 2000*). Briefly, SU-8 2050 molds containing circles of 250 µm in diameter were prepared by standard photolithography. Uncured PDMS was then spin-coated on molds to a thickness lower than the height of the microstructures (50 µm) and cured overnight at 65°C. Stencils for the finger experiments were prepared by spin-coating uncured PDMS on a flat surface.

### Cell seeding on stencils

The PDMS stencils were cut, peeled off the mold, and placed in a 2% Pluronic F-127 (Sigma-Aldrich) in PBS for 1 hr. The stencils were then kept in PBS for 2 hr and dried under the hood flow. PDMS stretchable membranes were cut and then activated using $O_2$ plasma. The membranes and the stencils were exposed to UV light in the cell culture hood for 10 min. Afterwards, stretchable membranes were incubated with fibronectin 20 µg/ml for 1 hr, rinsed with PBS and dried. PDMS membranes were placed on a PS holder, and the PDMS stencils were deposited on top of the membrane right after. A rectangular PDMS chamber was attached onto the membrane using vacuum grease, and cells were seeded at a density of 20,000 cells/mm$^2$ (*Serra-Picamal et al., 2012*) for 4 hr. When cells were attached, the medium was changed and the membrane with the cells was kept in the incubator. Local cell density could vary within each colony. We followed the dynamics of assembly of the colony prior removal of the stencil and we could see that cellular clusters size distribution and respective location within the cavity at plating could contribute to these variations. Once they formed confluent circular colonies, the stencils were removed with tweezers carefully before starting the experiment. Some of the colonies exhibited elongation in the short time window between stencil removal and the start of image acquisition.

### Time-lapse microscopy

After stencil removal, the medium was replaced by L-15 (Leibovitz's L-15 medium, Invitrogen) supplemented with 10% FBS. Cells were then observed under a Nikon Ti inverted microscope using either a x10 or a x20 objective for 6 hr at 37°C. Images were acquired every 5 min.

### Stretching experiments

The stretching device was designed in the lab. Briefly, a Thorlabs motor (Thorlabs z812 motor, Thorlabs) was controlling the motion of a PDMS membrane, and everything was mounted on a custom-made microscope stage. Circular colonies were plated on PDMS membranes, and after removal of the stencils, samples were placed on the microscope. Cyclic uniaxial stretches were applied and images were taken every 30 min typically shortly to prevent interfering with the time course of the

experiments. We probed three times for cycles, 20, 60, 120 s, and three extensions, 5, 10, and 20%. The shape of the cycles was triangular. We checked that the PDMS membranes were elastic at all extension and frequency ranges.

### Chemical treatments

To prevent the formation of E-cadherin-mediated adhesion, cells were incubated for 30 min with L-15 medium containing 5 mM EDTA (Sigma-Aldrich) and 10 µg/ml anti-E-cadherin blocking antibody that targeted the extracellular domain of E-cadherin (*Gumbiner et al., 1988*) (uvomorulin, monoclonal anti-E- cadherin antibody, Sigma); after incubation, the medium was replaced by normal L-15 and the experiment started. The inhibition of ROCK was done by incubating cells with Y-27632 50 µM solution (Sigma-Aldrich) from 30 min before the experiment started until the end of the experiment.

### Finger dynamics experiments

For the finger test after growth, we let finger grow for 2 hr, and we subsequently applied the cyclic stretch.

### Colony shape change analysis

Shape change analysis was performed using ImageJ (http://rsb.info.nih.gov/ij, NIH). The outline of the colony on phase contrast images was ellipse fitted at every time point. Major axis $a$, minor axis $b$, and ellipse orientation $\theta$ were obtained. We computed $Q$, defined as $Q_{xx}(t) = \frac{1}{2}\ln(a(t)/b(t))\cdot\cos2\cdot(\theta(t) - \alpha)$ and $Q_{xy}(t) = \frac{1}{2}\ln(a(t)/b(t))\cdot\sin2\cdot(\theta(t) - \alpha)$ being $\alpha = \theta(t_{final})$ to quantify cell colony elongation. In uniaxial stretching experiments, the $x$ axis corresponds to the direction of the external stretch and $Q$ components are defined as $Q_{xx}(t) = \frac{1}{2}\ln(a(t)/b(t))\cdot\cos2\cdot(\theta(t) - \alpha)$ and $Q_{xy}(t) = \frac{1}{2}\ln(a(t)/b(t))\cdot\sin2\cdot(\theta(t) - \alpha)$ being $\alpha = 0$.

### Velocity analysis

The centroid trajectories of cells were tracked using the manual tracking plug-in in ImageJ. Data analysis was performed using a custom-made code in MATLAB (The MathWorks). Cell positions were characterized by a vector $\mathbf{r}(t)$, with t denoting time and $\mathbf{r}$ position in space (bold letter refers to a vector). Every recorded cell position during the time-lapse experiment was defined as $\mathbf{r}_i = \mathbf{r}(t_i)$, where $t_i = i\Delta t$ are the times of recording and $\Delta t$ denotes the duration of time-lapses. The average velocity of each cell was then defined as $v = (1/T)\cdot\sum_i r_i$, being $r_i$ the module of the vector $\mathbf{r}_i$ and T the total duration of the trajectory.

### Colony segmentation and cell tracking

Movies acquired using an MDCK GFP-E-cadherin strain were first pre-processed with FIJI. The *subtract background* function was applied to remove noise. Images were then loaded to Tissue Analyzer (TA) v8.5 (*Aigouy et al., 2010*) for edge detection and cell tracking.

### Orientation field of the cells and topological defects

First, the background noise of the time-lapse images of the elongating MDCK colonies was reduced with the *subtract background* function by using a rolling ball radius of 40 px. The resulting images were then subjected to *band-pass* filter with upper and lower limits of 40 px and 3 px, respectively. The background noise from this output was reduced by using the *subtract background* command again with a rolling ball radius of 40 px. The processed images from each experiment were analyzed with the OrientationJ plugin of FIJI to quantify their local spatial orientation that reflects the underlying cell elongation. Within this plugin, we used a local smoothing window of 20 px (approximately of the size of the cells) to obtain the *structure tensor* at discrete points on a grid of 20 px × 20 px. The plugin provides the dominant direction $\phi_i$ of the structure tensor at each point $(x_i, y_i)$ that represents the local *orientation* field $\mathbf{q}_i = \cos\phi_i\hat{\mathbf{e}}_1 + \sin\phi_i\hat{\mathbf{e}}_2$ for cell elongation. The OrientationJ plugin also provides the *coherence* $C$ of the structure tensor to quantify the strength of the orientation; $C \approx 0$ and $C \approx 1$ would approximately correspond to rounded and elongated cells, respectively. The orientation or the director field $\mathbf{q}$ thus obtained was further quantified by studying the spatiotemporal evolution of $\pm 1/2$ topological defects that were obtained by calculating the winding number over unit-cells of

the underlying grid. The local smoothing window of 20 px for obtaining the *structure tensor*, which is approximately of the size of cells, ensured that the most robust defects were observed. The validity of this procedure was cross-verified with the smoothed cell-shape nematic field and the corresponding ±½ defects from the segmented and triangulated data of the experiments processed in Tissue Miner (TM) (*Figure 5—figure supplement 2* and Appendix 1). Finally, the orientation of mean cell-shape nematic calculated at 0 hr was compared with shape orientation of the colony at *t* = 2 hr. For obtaining the cell-shape nematic field for MCF 10A colonies, the procedure was the same as for MDCK control but the numerical parameters used were, rolling ball radius for *subtract background* 50 px, no band pass filter, and local smoothing window of 15 px and grid size of 30 px for OrientationJ. Similarly, for obtaining the cell-shape nematic field for stretched colonies, the procedure was the same as for MDCK control but the numerical parameters used were, rolling ball radius for *subtract background* 50 px, no band pass filter, and local smoothing window of 40 px and grid size of 30 px for OrientationJ. See Appendix 1 for more details (also see *Video 6*).

## Acto-myosin cables

In order to image acto-myosin supracellular cables at the boundary of colonies either cells expressing mCherry-actin / GFP-myosin strain or cell immunostaining were used. To disrupt acto-myosin cables, local injection of 4 µM Cytochalasin D using a micropipette was performed. Glass micropipettes were connected to a microinjection system (CellTram vario, Eppendorf). The position of the pipette tip was controlled in *x, y, z* by using a micromanipulator. The system was mounted on an epifluorescence inverted microscope to record the process. Cytochalasin D was released locally for about 10 min. To detect discontinuities in the cable, fluorescent images of actin and myosin were treated using Fiji as follows. First, a median filter was applied and the background subtracted. Then, a contrast-limited adaptive histogram equalization (CLAHE) was used and the background was again removed using an exponential function. Images corresponding to actin and myosin were added. Finally, a Laplacian of Gaussian filter was applied to the resulting image. Once the cable structure was revealed, the positions of the defects were identified.

## Quantification of cellular deformations, topological transitions, and their contribution to pure shear deformation

After obtaining the geometrical and topological information of the colonies from the series tracked images generated using TA, TM was used to extract, triangulate and store the data with the help of an automated workflow. The database obtained after this stage of analysis was used to quantify various state properties such as cell area, neighbor number, cell elongation and the contribution of different cellular processes to deformation using scripts written both in *R* and in *Python*. TM was further used to quantify the contributions of various cellular events such as cell elongation and topological transitions to the colony deformation. More details about this analysis can be found in Appendix 1 (also see *Video 7*).

## Vertex model simulations

A vertex model was developed with an addition of a unit nematic director **p** to every cell. The orientation of the boundary cell **p** was maintained parallel to the boundary, whereas **p** for internal cells were modeled to tend to align with the **p** of its neighbors. In these simulations, an extensile active stress $\sigma_a(\mathbf{pp} - \frac{1}{2}\mathbf{I})$ with $\sigma_a < 0$ acts to increase cell elongation along **p**. In addition, a bias λ was also applied on the basic edge tension with respect to the director **p** of its neighboring cells. For positive λ, this bias reduces (increases) the tension of the edge along (perpendicular to) **p**. Consequently, the closure (formation) of edges is enhanced in the direction perpendicular (parallel) to **p**. Hence, the T1 transitions in the region around the cells are oriented to cause shear elongation (contraction) along (perpendicular) to **p**. The colony is provided with an initial condition for **p** that mimics the initial configuration of experimentally frequently observed cell shape nematic fields **q** with two +½ defects that are separated by a distance. The initial polar vector orientation along the nematic axis are chosen at random such that the total polarity is zero, and the dynamics of polarity reorientation is independent on a **p** → -**p** flipping of the polarity axis. To begin with, the cell positions and director orientations were evolved under colony confinement until cell position and **p** do not change significantly. The confinement is then removed to see how the colony breaks symmetry in its shape. In

another set of simulations, a small motility $v_0$ was also provided to the internal cells (*Figure 7—figure supplement 1*). Similar to the experiments, the output of these simulations was also processed in TM and analyzed for topological defects and pure shear decomposition. For colonies subjected to uniaxial cyclic stretching, **p** for any cell was provided with an additional tendency to align along the stretching orientation (*x*-axis). Moreover, the active stress in this case $\sigma_a > 0$ was contractile, that is the cell tended to elongate perpendicular to the orientation of **p**. More details of the simulations are provided in Appendix 2 (also see *Videos 8* and *9*).

## Statistics

No statistical methods were used to predetermine sample size. Measurements were performed on different colonies (n) obtained in different independent experiments (N). The exact values are defined at the caption of each figure. Data presentation (as Mean value ± standard deviation (SD) or as Mean value ± standard error of the mean (SE)) is defined at the corresponding figure caption. D'Agostino normality test, Mann-Whitney test and Pearson's correlation were performed using GraphPad Prism 8. Specific values are noted at the corresponding figure captions.

## Acknowledgements

We acknowledge M Popovic and R Etournay for stimulating discussions and help with Tissue Miner, J Nelson, P Silberzan, S Coscoy for sharing the MDCK strains, and J Prost, K Dayal and M Zapotocky for helpful suggestions, D Rodriguez, M Lieb and the Riveline Lab. DR and MMI thank MPI-PKS and MPI-CBG for hospitality and many interactions. DR acknowledges support from CNRS (ATIP), ciFRC Strasbourg, the University of Strasbourg, Labex IGBMC, Foundation Cino del Duca. This study with the reference ANR-10-LABX-0030-INRT has been also supported by a French state fund through the Agence Nationale de la Recherche under the frame programme Investissements d'Avenir labelled ANR-10-IDEX-0002–02. MMI acknowledges funding from Department of Biotechnology, Government of India (BT/06/IYBA/2012) and Industrial Research and Consultancy Center (IRCC), IIT Bombay. GS was supported by the Francis Crick Institute which receives its core funding from Cancer Research UK (FC001317), the UK Medical Research Council (FC001317), and the Wellcome Trust (FC001317).

## Additional information

### Funding

| Funder | Grant reference number | Author |
| --- | --- | --- |
| Department of Biotechnology , Ministry of Science and Technology | BT/06/IYBA/2012 | Mandar M Inamdar |
| Industrial Research and Consultancy Centre | IRCC Awards, IIT Bombay | Mandar M Inamdar |
| Cancer Research UK | FC001317 | Guillaume Salbreux |
| Medical Research Council | FC001317 | Guillaume Salbreux |
| Wellcome Trust | FC001317 | Guillaume Salbreux |
| Centre National de la Recherche Scientifique | ANR-10-LABX-0030-INRT | Daniel Riveline |
| Agence Nationale de la Recherche | ANR-10-IDEX-0002-02 | Daniel Riveline |
| ciFRC Strasbourg | | Daniel Riveline |
| University of Strasbourg | | Daniel Riveline |
| ISIS | | Daniel Riveline |

The funders had no role in study design, data collection and interpretation, or the decision to submit the work for publication.

## Author contributions

Jordi Comelles, Conceptualization, Data curation, Formal analysis, Validation, Investigation, Visualization, Methodology, Writing - original draft, Writing - review and editing; Soumya SS, Data curation, Software; Linjie Lu, Emilie Le Maout, Investigation; S Anvitha, Software, Formal analysis; Guillaume Salbreux, Conceptualization, Software, Formal analysis, Investigation, Methodology, Writing - review and editing; Frank Jülicher, Conceptualization, Investigation, Writing - review and editing; Mandar M Inamdar, Conceptualization, Data curation, Software, Formal analysis, Supervision, Investigation, Visualization, Methodology, Writing - original draft, Writing - review and editing; Daniel Riveline, Conceptualization, Supervision, Funding acquisition, Validation, Investigation, Methodology, Writing - original draft, Project administration, Writing - review and editing

## Author ORCIDs

Jordi Comelles (iD) https://orcid.org/0000-0002-9297-830X
Soumya SS (iD) https://orcid.org/0000-0001-6141-5224
Guillaume Salbreux (iD) http://orcid.org/0000-0001-7041-1292
Frank Jülicher (iD) https://orcid.org/0000-0003-4731-9185
Mandar M Inamdar (iD) https://orcid.org/0000-0001-8549-8490
Daniel Riveline (iD) https://orcid.org/0000-0002-4632-011X

## Decision letter and Author response

Decision letter https://doi.org/10.7554/eLife.57730.sa1
Author response https://doi.org/10.7554/eLife.57730.sa2

---

# Additional files

## Supplementary files

• Transparent reporting form

## Data availability

All data generated or analysed during this study are included in the manuscript and supporting files.

---

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

# Appendix 1

## Quantification of tissue kinematics

In the first step of analysis, time-lapse images of MDCK colonies are segmented to partition the tissue into individual cells and then tracked for their movements using Tissue-Analyser (TA) plugin of FIJI (*Aigouy et al., 2010*; *Aigouy et al., 2016*) (see *Figure 6—figure supplement 1a-i*). This information of the tracked cell colony is then passed to Tissue-Miner (TM), which extracts data and stores the cellular geometry and connectivity in the form of cell vertices and bonds using a *snakemake* based automated workflow (see *Figure 6—figure supplement 1j-l*). As described in *Merkel et al., 2017*; *Etournay et al., 2016*, TM has *R* or *Python* API that can be used to quantify fine-grained (cell-level) and coarse-grained (colony-level) deformation properties of epithelial colonies. Even though the scripts to calculate the basic kinematic quantities are available, users have the freedom to add new computational routines to calculate the quantities of their interest.

### A. Analysis of cell elongation

Cell-shape anisotropy is an important marker of stress in the cells in response to either internal activity or external forcing (*Paluch and Heisenberg, 2009*; *Rauzi and Lenne, 2011*). To understand and quantify the elongation of cells over time, TM computes an elongation parameter proposed in *Aigouy et al., 2010*. In their study to understand the role of flow in orienting the axis of polarity in *Drosophila* wings, *Aigouy et al., 2010* use a nematic tensor with a magnitude and orientation to quantify the elongation of each cell. For a given cell with area $A_c$, the elongation nematic tensor is proposed as

$$E = \begin{bmatrix} \epsilon_{xx} & \epsilon_{xy} \\ \epsilon_{xy} & -\epsilon_{xx} \end{bmatrix}. \tag{1}$$

This tensor $E$ is traceless and symmetric with components given by

$$
\begin{aligned}
\epsilon_{xx} &= \frac{1}{A_c} \int_0^{2\pi} \int_0^{R(\phi)} \cos(2\phi) r \, dr \, d\phi \\
\epsilon_{xy} &= \frac{1}{A_c} \int_0^{2\pi} \int_0^{R(\phi)} \sin(2\phi) r \, dr \, d\phi,
\end{aligned}
\tag{2}
$$

where $R(\phi)$ is the distance to cell boundary at an angle $\phi$ (SI of *Aigouy et al., 2010*). For this nematic tensor, the magnitude is given by

$$\epsilon = (\epsilon_{xx}{}^2 + \epsilon_{xy}{}^2)^{\frac{1}{2}}, \tag{3}$$

and the orientation is given by angle $\phi$

$$
\begin{aligned}
\cos(2\phi) &= \frac{\epsilon_{xx}}{\epsilon}, \\
\sin(2\phi) &= \frac{\epsilon_{xy}}{\epsilon}.
\end{aligned}
\tag{4}
$$

Based on the above expressions, the spatial distribution of the magnitude of shape nematic for individual cells within the colony for three time points is shown in *Figure 6—figure supplement 2a-c*. The fine-grained nematic segments for individual cells whose magnitude is proportional to $\epsilon$ and orientation is $\phi$ (*Figure 6—figure supplement 2d-f*). The coarse-grained representation of this tensor (*Figure 6—figure supplement 2g-i*) is obtained by spatially averaging the elongation nematic over several cells by using a Gaussian smoothing kernel of size comparable to the size of cells (30 pixels in this case). From the figures it is clear that the elongation of interior cells is strongly correlated with the orientation of the future 'finger' even at $t = 0$ h.

## B. Analysis of tissue deformation from components of pure shear strain via triangulation

As discussed earlier, during different stages of morphogenesis, epithelial tissues are reported to undergo a series of physical changes. The various internal and external forces that are being applied to the tissue during these processes result in deformation and rearrangement of constituent cells. The shape modification rate of the tissue is quantified in terms of the anisotropic component of shear strain rate. However, at the tissue level in epithelial sheets, shear strain can arise due to a combination of cell elongation, T1 transitions, cell division, cell extrusion, and correlation effects. By studying each both component separately and with respect to the other components, one can get hints regarding the mechano-chemical state of the colony. As is described in *Etournay et al., 2016*, for any general 2-D velocity vector $\mathbf{v}$, the gradient tensor can be defined as

$$v_{ij} = \partial_i v_j = \begin{bmatrix} \frac{\partial v_x}{\partial x} & \frac{\partial v_y}{\partial x} \\ \frac{\partial v_x}{\partial y} & \frac{\partial v_y}{\partial y} \end{bmatrix}. \tag{5}$$

The above velocity gradient tensor for a group of cells can be split into three parts: (i) isotropic component to obtain the rate of area change, (ii) symmetric traceless component to quantify the pure shear strain rate, and (iii) an antisymmetric part that represents the local vorticity (*Etournay et al., 2015*; *Etournay et al., 2016*) as follows

$$v_{ij} = \frac{1}{2} v_{kk} \delta_{ij} + \tilde{v}_{ij} + \omega \delta_{ij}$$
$$= \frac{1}{2} \underbrace{\begin{bmatrix} \frac{\partial v_x}{\partial x} + \frac{\partial v_y}{\partial y} & 0 \\ 0 & \frac{\partial v_x}{\partial x} + \frac{\partial v_y}{\partial y} \end{bmatrix}}_{v_{kk}:\text{isotropic}} + \frac{1}{2} \underbrace{\begin{bmatrix} \frac{\partial v_x}{\partial x} - \frac{\partial v_y}{\partial y} & \frac{\partial v_y}{\partial x} + \frac{\partial v_x}{\partial y} \\ \frac{\partial v_y}{\partial x} + \frac{\partial v_x}{\partial y} & \frac{\partial v_y}{\partial y} - \frac{\partial v_x}{\partial x} \end{bmatrix}}_{\tilde{v}_{ij}:\text{symmetric,anisotropic}} + \frac{1}{2} \underbrace{\begin{bmatrix} 0 & \frac{\partial v_y}{\partial x} - \frac{\partial v_x}{\partial y} \\ \frac{\partial v_x}{\partial y} - \frac{\partial v_y}{\partial x} & 0 \end{bmatrix}}_{\omega:\text{antisymmetric}}.$$

Here, more specifically, $v_{kk} = v_{xx} + v_{yy}$ is the trace of the velocity gradient tensor, or the divergence of the velocity field and quantifies the rate of contraction or expansion in the cell area. The traceless symmetric part $\tilde{v}_{ij}$ is the pure shear strain rate and quantifies shape changes in the tissue. The antisymmetric component $\omega$ is the vorticity of the velocity field and provides values of rotational rates of the group of cells during deformation.

The total pure shear strain rate of the tissue $\tilde{v}_{ij}$, now represented in a tensorial form $\tilde{\mathbf{V}}$, can be further split into traceless, symmetric, components as follows

$$\tilde{\mathbf{V}} = \frac{D\mathbf{Q}}{Dt} + \mathbf{T} + \mathbf{C} + \mathbf{E} + \mathbf{D}, \tag{7}$$

where $\mathbf{Q}$ is the shear contribution coming from average cell elongation, $\mathbf{T}$ is the shear contribution from T1 transitions, $\mathbf{C}$ from cell divisions, $\mathbf{E}$ from cell extrusions and $\mathbf{D}$ due to the correlations effects in the tissue which arise due to coarse graining effects. Detailed derivation and discussion of this expression are available in *Etournay et al., 2015*; *Etournay et al., 2016*; *Merkel et al., 2017*. The contributions from above mentioned components are quantified using the triangulation approach (*Merkel et al., 2017*). As shown in *Figure 6—figure supplement 2j-l*, the triangulation method considers the entire colony without the boundary cells as a network of triangles, connecting the centers of the neighboring cells. Any topological changes in the form of cell division or extrusion will lead to a new triangulation. The shear strain rates in *Equation 7* can be integrated over time to provide cumulative shape deformation and the corresponding cumulative shear decomposition for the cell colony, and is obtained for the complete colony without including the external boundary cells (*Figure 6b*). Figure 4 of *Etournay et al., 2015* schematically represents this decomposition of total tissue shear into individual cellular events.

In *Drosophila* pupal wing the *x* axis is naturally provided by the A-P axis. In our in vitro situation, however, the *x* axis is chosen as the dominant direction of colony elongation, such that at the end of the experiment ($t = 2$ h) the cumulative value of the total shear strain rate $\tilde{v}_{xy} = 0$. The cumulative shear decomposition plots for the experiments and simulations are shown in *Figure 6*, *Figure 6—figure supplement 3* and *Figure 7—figure supplement 1*.

## C. Correlation between the orientation of mean cell-shape nematic and the orientation of the overall colony elongation

As discussed earlier, the shape of a cell can be thought of to be represented by a nematic tensor $\mathbf{Q}^{\text{cell}}$. Upon averaging this quantity over multiple cells, we obtain nematic field $\mathbf{Q}^{\text{nem}}(x, y, t)$ over the colony. As described in the previous sub-section, this nematic tensor is traceless and symmetric with an orientation $\phi(x, y, t)$ (*Figure 5—figure supplement 2* and *Figure 6—figure supplement 2g-i*) and magnitude $Q^{\text{nem}}(x, y, t)$ such that

$$Q_{xx}^{\text{nem}}(x, y, t) = Q^{\text{nem}} \cos 2\phi \text{ and} \tag{8}$$

$$Q_{xy}^{\text{nem}}(x, y, t) = Q^{\text{nem}} \sin 2\phi. \tag{9}$$

This nematic field is obtained for our MDCK colonies by analyzing time-lapse images with the OrientationJ plugin of ImageJ (*Püspöki et al., 2016*; *Saw et al., 2017*). We verified that this image based approach gives results that are comparable with those obtained by spatially smoothing (with Gaussian kernel) the individual cell-shape nematics extracted from the segmented images of the colonies (Appendix 1A, *Figure 5—figure supplement 2* and *Figure 6—figure supplement 2*). At any particular time $t$, the mean nematic field for the colony is

$$Q_{xx}^{\text{ncol}}(t) = \langle Q_{xx}^{\text{nem}}(x, y, t) \rangle_{x,y} \text{ and} \tag{10}$$

$$Q_{xy}^{\text{ncol}}(t) = \langle Q_{xy}^{\text{nem}}(x, y, t) \rangle_{x,y}. \tag{11}$$

The orientation of the mean cell-shape nematic is then simply

$$\theta_{\text{nematic}}(t) = \frac{1}{2} \tan^{-1} \left( \frac{Q_{xy}^{\text{ncol}}(t)}{Q_{xx}^{\text{ncol}}(t)} \right), \tag{12}$$

where $\tan^{-1}$ is implemented in Matlab using the inbuilt function atan2 (*MATLAB, 2018*). In this analysis, the *coherence* of the image structure tensor is the counterpart of the magnitude of the mean cell-shape nematic (*Püspöki et al., 2016*). We then compare the final colony orientation $\theta_{\text{colony}}(t = 2 \text{ hrs})$, which is obtained from ellipse fitting, with the orientation of the mean colony nematic $\theta_{\text{nematic}}(t = 0 \text{ hrs})$ (*Figure 5b* and *Figure 5—figure supplement 2*). Also refer to Materials and Methods: Orientation field of the cells and topological defects in the Main Paper for additional details.

## D. Topological defects as readouts of cell-shape nematic orientation field

Any nematic field can contain topological defects (*Kleman and Laverntovich, 2003*). A visual inspection of the cell-shape orientation field $\mathbf{q} = \cos \phi \hat{\mathbf{e}}_x + \sin \phi \hat{\mathbf{e}}_y$ for MDCK colonies indicates the possible presence of topological defects (*Figure 5* and *Figure 5—figure supplement 2*). The strength $k$ of topological charge for this field (Appendix 1C) in a given region is obtained as

$$k = \frac{1}{2\pi} \oint \frac{d\phi}{ds} ds, \tag{13}$$

where the line integral is calculated by traversing the curve that encloses this region in an anticlockwise sense. For the experimental colonies, the nematic field is not continuous but obtained on the vertices of the grid-cell as shown in *Figure 5—figure supplement 5*. In this case, the topological charge $k$ enclosed within the grid-cell is

$$k = \frac{1}{2\pi} \sum_j \Delta\Phi_{j+1,j}, \tag{14}$$

where the notation is as shown in *Figure 5—figure supplement 5*. If no topological defect is present within the cell then $k = 0$. Else, the net topological charge would evaluate to $k = \pm 1/2$, corresponding to the effective strength of the topological defects within the cell.

The location and strength of topological defects thus obtained for the experimental data could then be used to approximately depict the entire nematic field as follows. We first model the energetics of the nematic orientation field with one constant Frank free energy (*de Gennes and Prost, 1995*). If we then assume that the orientation field $\phi$ is in local mechanical equilibrium at every time-step, it will then satisfy the Laplace equation $\nabla^2\phi = 0$ (*Kleman and Laverntovich, 2003*). The resulting orientation field $\phi(x, y)$ for the nematic field in the presence of a single topological defect of strength $k$ at the origin has a singular solution of the form

$$\phi = A\theta + \alpha, \tag{15}$$

where $\theta$ is the polar angle and $\alpha$ is a harmonic function that satisfies that Laplace equation (*Kleman and Laverntovich, 2003*). However, for the current paper we find that taking $\alpha$ to be a constant that determines the orientation of the topological defect is sufficient to approximate the orientation of nematic field (*Figure 5d*). Hence, as per *Equations 13 and 15* the integral

$$\oint d\phi = 2\pi k, \tag{16}$$

for a closed curved around the defect gives $A = k$. If multiple defects are present in the domain, this solution for $\phi$ can be generalised to give

$$\phi_\alpha(x, y) = \alpha + \sum_i k_i \underbrace{\tan^{-1}\left(\frac{y - y_i}{x - x_i}\right)}_{\theta \text{ for defect } i}, \tag{17}$$

where $\{x_i, y_i\}$ and $k_i$ are the coordinates and the strength, respectively, of the $i^{\text{th}}$ nematic topological defect obtained from the experimental data, and $\alpha$ dictates the orientation of the defect. Here, the subscript $\alpha$ in $\phi_\alpha(x, y)$ is added to distinguish it from the experimentally obtained $\phi(x, y)$. As before, $\tan^{-1}$ is implemented in Matlab using the inbuilt function atan2 (*MATLAB, 2018*). We now use the fitting parameter $\alpha$ in $\phi_\alpha(x, y)$ to see how well it represents the orientation field $\phi(x, y)$ by using the following procedure. The experimental field is obtained on a grid with $N$ points $\{x_k, y_k\}$, where $k = 1, \ldots, N$. The difference between $\phi$ and $\phi_\alpha$ is quantified by the residue

$$r_\alpha = -\frac{1}{N}\sum_{k=1}^{N} \cos^2(\phi^k - \phi_\alpha^k) \tag{18}$$

obtained over the grid points $k$. The minimization of the value of $r_\alpha$ over $\alpha$ would provide us with a field where $\phi \approx \phi_\alpha$ in a mean sense. Interestingly, by using this simple approach with just one fitting parameter $\alpha$, we can get an excellent representation of the experimentally obtain nematic orientation field $\hat{\mathbf{q}}$ (*Figure 5d*).

## E. Statistical test for cumulative distribution of the orientation differences between $\theta_{\text{nematic}}(0 \text{ hrs})$ and $\theta_{\text{colony}}(2 \text{ hrs})$

To check if the orientation of the colony elongation at $t = 2 > \text{hrs}$ is correlated with the orientation of the mean cell-shape nematic at $t = 0 > \text{hrs}$, we examine the difference $\Delta\theta = |\theta_{\text{colony}}(2 > \text{hrs}) - \theta_{\text{nematic}}(0 > \text{hrs})|$. Specifically, as shown in *Figure 5c*, we obtain the cumulative distribution function (CDF) for $\Delta\theta$. If the quantity $\Delta\theta$ is completely random, i.e., $\theta_{\text{colony}}(2 > \text{hrs})$ is completely uncorrelated with $\theta_{\text{nematic}}(0 > \text{hrs})$, then the probability density for $\Delta\theta \in [0°, 90°]$ is equal to $1/90°$. In such a case the CDF($\Delta\theta$) as a function of $\Delta\theta$ should simply be a straight line with a slope of $1/90°$. However, it can clearly be seen that the experimentally observed CDF is significantly above this line, especially at relatively smaller values of $\Delta\theta \leq 30°$. This indicates that there indeed is correlation between $\theta_{\text{colony}}(2 > \text{hrs})$ and $\theta_{\text{nematic}}(0 > \text{hrs})$.

We now note that the total number of experiments (colonies) that were analysed for *Figure 5c* is $N = 19$. To check if the correlation between $\theta_{\text{colony}}(2 > \text{hrs})$ and $\theta_{\text{nematic}}(0 > \text{hrs})$ holds up for this finite sample size, we do the following. Let us take the probability density $p(\Delta\theta) = 1/90°$, $> \forall \Delta\theta \in [0°, 90°]$, i.e., this relative angle $\Delta\theta$ is uniformly random between $0°$ and $90°$. If we now sample $N = 19$ values for $\Delta\theta$ then we have created a *random* counterpart to the actual experimental result. We repeat this

trial using a straightforward Matlab script and obtain $M = 10^6$ samples, each with $N = 19$ experiments. Of all the samples thus randomly generated, for a given $\Delta\theta$ we select those samples $i$ for which $\mathrm{CDF}_i^{\mathrm{random}}(\Delta\theta) \geq \mathrm{CDF}^{\mathrm{experiment}}$. If for a particular $\Delta\theta$ the total number of such selected samples is $m(\Delta\theta)$, the probability of observing equal or higher bias towards alignment of the two angles from a random experiment than that actually observed experimentally is $P(\Delta\theta) = m(\Delta\theta)/M$ (**Figure 5—figure supplement 2d**). It can clearly be seen that the percentage probability of observing such high values of CDF for smaller $\Delta\theta < 30°$ from a completely random sampling of $\Delta\theta$ is $\approx 1$, i.e., approximately one sample of $N = 19$ experiments for every 100 samples, a very low number. The few colonies for which $\theta_{\mathrm{nematic}}(0\ \mathrm{hr})$ does not match well with $\theta_{\mathrm{colony}}(2\ \mathrm{hr})$ appear to align with the orientation $\theta_{\mathrm{colony}}(0\ \mathrm{hr})$ of tissue initial elongation – the associated mechanism is, however, still open (see **Figure 5—figure supplement 2e**). This entire analysis clearly indicates that the orientation of the mean cell-shape nematic at the initial time has correlation with the final anisotropy orientation of the colony with a high probability.

## Appendix 2

### Vertex model simulation

A planar vertex model is one of the most commonly used computational tool to understand mechanics of epithelial monolayers (*Farhadifar et al., 2007*). In this description, the tissue is assumed to be a 2-D sheet composed of cells which are represented by planar polygons sharing vertices and edges with their neighbors (*Figure 7a*). Governing equations are formulated to study the dynamics of evolution of each vertex of a cell. The total effective *energy* or work-function $W$ of the tissue arises from area deformation of the cells and a combination of acto-myosin contractility and membrane adhesion energy at the cell interfaces (*Figure 7a*).

$$W = \sum_{\alpha=1}^{N} \frac{K_\alpha}{2}(A_\alpha - A_\alpha^0)^2 + \sum_{<\alpha\beta>} \Lambda_{\alpha\beta} l_{\alpha\beta}, \tag{19}$$

where for a particular cell $\alpha$, $A_\alpha$ is the actual area and $A_\alpha^0$ is the preferred cell area. $l_{\alpha\beta}$ is the length of the bond connecting shared between the cells $W$ and $\beta$ and $\Lambda_{\alpha\beta}$ is the effective contractility of that bond (*Alt et al., 2017*). Note that the energy $W$ depends both on the position $\mathbf{r}_i$ of the vertices and cell connectivity.

For this model, the force $\mathbf{F}_{\text{vm}}^i$ acting on a particular vertex $i$ can be obtained by taking a derivative of $W$ (*Fletcher et al., 2014*; *Farhadifar et al., 2007*) with respect to the vertex position $\mathbf{r}_i$ as,

$$\mathbf{F}_{\text{vm}}^i = -\frac{\partial W}{\partial \mathbf{r}_i} \tag{20a}$$

$$= \eta \frac{d\mathbf{r}_i}{dt}, \tag{20b}$$

and is balanced by the friction force on the vertex from the substrate. Here, $\eta$ represents effective substrate viscosity experienced by the vertex via cell-substrate friction (*Fletcher et al., 2014*). A mathematically equivalent description of the vertex model could also be made by using mechanical equilibrium or principle of virtual work (*Chen and Brodland, 2000*; *Alt et al., 2017*).

### A. Inclusion of active cell stress in the vertex model formulation

Collectively migrating epithelial cells can have an active stress of the form (*Popović et al., 2017*)

$$\sigma = \sigma_a(\hat{\mathbf{p}}\hat{\mathbf{p}} - \frac{1}{2}\mathbf{I}), \tag{21}$$

where for $\sigma_a > 0$ and $\sigma_a < 0$, respectively, for contractile and extensile cell active stress. When $\sigma_a > 0$, the cell has a tendency to contract along its polarisation $\hat{\mathbf{p}}$ and elongate perpendicular it, and vice-versa when $\sigma_a < 0$. The goal of this section is to present how this stress within a particular cell can be accommodated in the vertex model formalism by obtaining the corresponding active force $\mathbf{F}^i$ on the vertex $i$ of the cell (*Figure 7a*).

### 1. Formulation of obtaining virtual strain of the cell from virtual displacement of cell vertices

We note that, when the vertices are provided virtual displacement of $\{\delta x_\alpha^i\}$, the total internal virtual work because of the action of the cell active stress $\sigma^\alpha$ on the resulting virtual deformation gradient $\delta\mathbf{u}$ is

$$\delta W_{\text{active}} = A\sigma_{\alpha\beta}\delta u_{\alpha\beta} = -\sum_i F_\alpha^i \delta x_\beta^i, \tag{22}$$

where $A$ is the instantaneous area of the cell. We note that the negative sign in front of the summation sign arises since $F_\alpha^i$ is taken as the force exerted by the interior of the cell on the vertices.

If we can find a relation between $\delta u_{\alpha\beta}$ and $\{\delta x_{\alpha}^i\}$, then **Equation 22** will provide us with expression for $F_{\alpha}^i$. We first note that, assuming an affine virtual displacement field, the virtual displacement $\delta x_{\alpha}^i$ of any vertex $i$ of a given cell $k$ will be given as

$$\begin{aligned} \delta x_{\alpha}^i &= \delta u_{\alpha\beta}(x_{\beta}^i - x_{\beta}^c) + \delta x_{\alpha}^c, \text{ and} \\ \delta \rho_{\alpha}^i &= \delta u_{\alpha\beta}\rho_{\beta}^i, \end{aligned}$$

(23)

where the superscript $c$ denotes the centroid of vertices, with position

$$x_{\alpha}^c = \frac{1}{N_{\text{ver}}}\sum_{i\in\text{cell}} x_{\alpha}^i,$$

(24)

the virtual displacement of this centroid is

$$\delta x_{\alpha}^c = \frac{1}{N_{\text{ver}}}\sum_{i\in\text{cell}} \delta x_{\alpha}^i,$$

(25)

and

$$\begin{aligned} \delta \rho_{\alpha}^i &= \delta x_{\alpha}^i - \delta x_{\alpha}^c, \\ \rho_{\alpha}^i &= x_{\alpha}^i - x_{\alpha}^c. \end{aligned}$$

(26)

Here, $N_{\text{ver}}$ is the number of vertices of the cell under consideration. The connection between $\delta\rho_{\alpha}^i$ and $\delta u_{\alpha\beta}$ as per **Equation 23** provides a set of $2\times(N_{\text{ver}}-1)$, independent linear equations, and can be guaranteed a unique solution in terms of the four independent components of the tensor $\delta u_{\alpha\beta}$, i. e., $\delta u_{11}, \delta u_{12}, \delta u_{21}$ and $\delta u_{22}$ only for $N_{\text{ver}}=3$. For $N_{\text{ver}}>3$, we obtain a solution in the sense of least squares. To do that we minimise the error

$$S = \sum_{i\in\text{cell}}(\delta\rho_{\alpha}^i - \delta u_{\alpha\beta}\rho_{\beta}^i)(\delta\rho_{\alpha}^i - \delta u_{\alpha\gamma}\rho_{\gamma}^i),$$

(27)

where repeated summation is implied over the Greek numerals $\alpha$, $\beta$ and $\gamma$, which can take values of 1 and 2. This error $S$ is to be minimised over $\delta u_{\mu\nu}$, and will give four independent linear equations. Differentiating $S$ with respect to $\delta u_{\mu\nu}$, we get the following set of equations (in indicial notation)

$$\frac{\partial S}{\partial \delta u_{\mu\nu}} = -2\sum_{i\in\text{cell}}(\rho_{\nu}^i\delta\rho_{\mu}^i - \delta u_{\mu\beta}\rho_{\beta}^i\rho_{\nu}^i) = 0.$$

(28)

These equations can be written in more readable format in the following manner.

$$\begin{bmatrix} \delta\rho_{11} \\ \delta\rho_{21} \\ \delta\rho_{12} \\ \delta\rho_{22} \end{bmatrix} = \begin{bmatrix} \rho_{11} & \rho_{12} & 0 & 0 \\ \rho_{12} & \rho_{22} & 0 & 0 \\ 0 & 0 & \rho_{11} & \rho_{12} \\ 0 & 0 & \rho_{12} & \rho_{22} \end{bmatrix} \begin{bmatrix} \delta u_{11} \\ \delta u_{12} \\ \delta u_{21} \\ \delta u_{22} \end{bmatrix}$$

(29)

where the quantities are defined as follows:

$$\begin{aligned} \rho_{\alpha} &= \sum_i \rho_{\alpha}^i = 0, \\ \rho_{\alpha\beta} &= \sum_i \rho_{\alpha}^i\rho_{\beta}^i, \\ \delta\rho_{\alpha\beta} &= \sum_i \rho_{\alpha}^i\delta\rho_{\beta}^i, \end{aligned}$$

(30)

such that $\alpha \in \{1,2\}$. **Equation 29** can be solved to provide the components of $\delta U$ as follows.

$$\delta u_{11} = \frac{\rho_{12}\delta\rho_{21} - \rho_{22}\delta\rho_{11}}{\rho_{12}^2 - \rho_{11}\rho_{22}},$$

$$\delta u_{12} = \frac{\rho_{12}\delta\rho_{11} - \rho_{11}\delta\rho_{21}}{\rho_{12}^2 - \rho_{11}\rho_{22}},$$

$$\delta u_{21} = \frac{\rho_{12}\delta\rho_{22} - \rho_{22}\delta\rho_{12}}{\rho_{12}^2 - \rho_{11}\rho_{22}},$$

$$\delta u_{22} = \frac{\rho_{12}\delta\rho_{12} - \rho_{11}\delta\rho_{22}}{\rho_{12}^2 - \rho_{11}\rho_{22}}.$$

(31)

Thus in *Equation 31* we have the virtual deformation tensor for a given cell $k$ in terms of the virtual displacement of its vertices $\{\delta\rho_\alpha^i\}$ with respect to its vertex centroid. These expressions are to be substituted in *Equation 22* to find the forces that are exerted on cell vertices due to the active cell stress in *Equation 21*.

## 2. Connection between cell active stress and vertex forces

The active stress for the cell in *Equation 21* can be written in indicial notation as

$$\sigma_{\alpha\beta} = \sigma_a(p_\alpha p_\beta - \frac{1}{2}\delta_{\alpha\beta}),$$

(32)

and the contribution to the virtual work is

$$\delta W_{\text{active}} = A\sigma_{\alpha\beta}\delta u_{\alpha\beta}.$$

(33)

Using the expression for $\delta u_{\alpha\beta}$ from *Equation 31* we get

$$\begin{aligned} A\sigma_{\alpha\beta}\delta u_{\alpha\beta} &= A\sigma_{11}\delta u_{11} + A\sigma_{12}(\delta u_{12} + \delta u_{21}) + A\sigma_{22}\delta u_{22} \\ &= A\sigma_a(p_1^2 - 1/2)(\frac{\rho_{12}\delta\rho_{21} - \rho_{22}\delta\rho_{11}}{\rho_{12}^2 - \rho_{11}\rho_{22}}) \\ &+ A\sigma_a(p_2^2 - 1/2)(\frac{\rho_{12}\delta\rho_{12} - \rho_{11}\delta\rho_{22}}{\rho_{12}^2 - \rho_{11}\rho_{22}}) \\ &+ A\sigma_a(p_1 p_2)(\frac{\rho_{12}\delta\rho_{11} - \rho_{11}\delta\rho_{21}}{\rho_{12}^2 - \rho_{11}\rho_{22}} + \frac{\rho_{12}\delta\rho_{22} - \rho_{22}\delta\rho_{12}}{\rho_{12}^2 - \rho_{11}\rho_{22}}), \\ &= -\sum_{\text{vertices } i} F_\alpha^i \delta x_\alpha^i. \end{aligned}$$

(34)

We now note from *Equation 30* that

$$\delta\rho_{\alpha\beta} = \sum_i \rho_\alpha^i(\delta x_\beta^i - \delta x_\beta^c) = \sum_i \rho_\alpha^i \delta x_\beta^i.$$

(35)

Using *Equation 35*, *Equation 34* is rewritten as

$$\begin{aligned} A\sigma_{\alpha\beta}\delta u_{\alpha\beta} &= A\sigma_a \sum_i [\frac{(p_1^2 - 1/2)[\rho_{12}\rho_2^i - \rho_{22}\rho_1^i] + (p_1 p_2)[\rho_{12}\rho_1^i - \rho_{11}\rho_2^i]}{\rho_{12}^2 - \rho_{11}\rho_{22}}\delta x_1^i \\ &+ \frac{(p_2^2 - 1/2)[\rho_{12}\rho_1^i - \rho_{11}\rho_2^i] + (p_1 p_2)[\rho_{12}\rho_2^i - \rho_{22}\rho_1^i]}{\rho_{12}^2 - \rho_{11}\rho_{22}}\delta x_2^i]. \end{aligned}$$

(36)

Hence, from *Equations 36 and 34* the contribution to the force on vertex $i$ due to the cell active stress from the cell is

$$\begin{aligned} F_1^i &= -A\sigma_a \frac{(p_1^2 - 1/2)[\rho_{12}\rho_2^i - \rho_{22}\rho_1^i] + (p_1 p_2)[\rho_{12}\rho_1^i - \rho_{11}\rho_2^i]}{\rho_{12}^2 - \rho_{11}\rho_{22}}, \\ F_2^i &= -A\sigma_a \frac{(p_2^2 - 1/2)[\rho_{12}\rho_1^i - \rho_{11}\rho_2^i] + (p_1 p_2)[\rho_{12}\rho_2^i - \rho_{22}\rho_1^i]}{\rho_{12}^2 - \rho_{11}\rho_{22}}. \end{aligned}$$

(37)

It can very easily be seen from *Equation 37* that for a given cell, the sum of all the active forces on the vertices of that cell will be zero. Moreover, the total torque of all these forces with respect to

the vertex centroid as defined in *Equation 24* can also be shown to be equal to zero. This is as expected from a symmetric stress tensor. In the current work, we take the cell active stress as extensile, and hence $\sigma_a < 0$ (*Saw et al., 2017*).

Finally, the contribution of this active force to any vertex is a vector sum of the force contributions coming from all the cells that the vertex is a part of, and is given by,

$$(F_\alpha^i)^{\text{total}} = \sum_{\text{cell } k \in \text{vertex } i} (F_\alpha^i)^{\text{total}}; \ \ \alpha \in \{1,2\}. \tag{38}$$

The active force on the vertex would be added to the forces from cell area deformation and cell edge contractility as discussed in the earlier section.

## B. Polarised or biased edge tensions

It is observed that T1 transitions can be influenced by the polarity of the cells (*Sato et al., 2015*). Such bias in T1 transitions can be introduced in the vertex model by modifying the tension $\lambda_{\alpha\beta}$ of any edge $\alpha\beta$ based on its relative orientation $\hat{\mathbf{e}}_{\alpha\beta}$ with respect to the polarity $\hat{\mathbf{p}}_\alpha$ and $\hat{\mathbf{p}}_\beta$ of the shared cells (*Figure 7a*) as

$$\Lambda_{\alpha\beta} = \Lambda_{\alpha\beta}^0 + \lambda(1 - [(\hat{\mathbf{e}}_{\alpha\beta} \cdot \hat{\mathbf{p}}_\alpha)^2 + (\hat{\mathbf{e}}_{\alpha\beta} \cdot \hat{\mathbf{p}}_\beta)^2]) + \xi(t), \tag{39}$$

where the quantities $\Lambda_{\alpha\beta}^0$, $\lambda$, $\xi(t)$ are, respectively, the base edge tension, angle dependent edge tension bias, and correlated noise that is calculated as

$$\frac{d\xi(t)}{dt} = -\frac{1}{\tau}\xi(t) + \gamma(t), \text{ such that} \tag{40}$$

$$\langle \gamma(t) \rangle = 0, \text{ and} \tag{41}$$

$$\langle \gamma(t')\gamma(t'+t) \rangle = \frac{2\Delta\Lambda^2}{\tau}\delta(t). \tag{42}$$

Here, $\tau$ and $\Delta\Lambda$ are the correlation time and strength, respectively, for the correlated noise. This noise promotes T1 transitions and introduces tissue fluidity in the vertex model (*Curran et al., 2016*). The angle dependent bias ensure that the edges that are parallel (perpendicular) to the polarisation of the cells have a tension that is lower (higher) by $\lambda$ than its base value $\Lambda_{\alpha\beta}^0$. This bias enhances the propensity of tissue shear along cell polarisation due to T1 transitions.

## C. Inclusion of cell motility

In addition to the forces described in the previous section, cells can also generate motility force (*Mogilner and Oster, 1996*; *Pollard and Borisy, 2003*). Following the idea of self-propelled cells (*Bi et al., 2016*), any cell $\alpha$ can also be modeled to have a speed $v_0$, such that in the absence of any other forces it moves with a velocity $v_0\hat{\mathbf{p}}_\alpha$, where $\hat{\mathbf{p}}_\alpha$ is the polarisation direction of the cell as described earlier (*Figure 7a*). A simplest representation of motile force on a particular vertex $i$ is

$$\mathbf{F}_{\text{motile}}^i = \eta v_0 \frac{1}{N_i} \sum_{\text{cell } \beta} \hat{\mathbf{p}}_\beta, \tag{43}$$

where $N_i$ is the total number of cells that contain the particular vertex $i$, and $\beta$ is the index of each of these cells (*Sussman, 2017*). The motile force, if included, will be in addition to the forces as described in the previous sections.

## D. Dynamical evolution

Dynamical evolution in these simulations involves updating the position of vertex $i$ as per the following equation

$$\eta \frac{d\mathbf{r}_i}{dt} = \mathbf{F}^i_{\mathrm{basic}} + \mathbf{F}^i_{\mathrm{active}} + \mathbf{F}^i_{\mathrm{motile}}, \tag{44}$$

where the subscripts basic, active, and motile correspond to the contributions to the forces, respectively, from basic vertex model, active stress and biased edge tensions, and cell motility. The evolution of edge tensions is as discussed in the previous section. In addition to the evolution of the vertex positions, whenever the size of any edge becomes smaller than a critical size $\epsilon_{\mathrm{cls}}$, a connectivity change, termed as T1 transition in the literature *Alt et al., 2017*; *Fletcher et al., 2014*, is implemented – the size of the new edge $\epsilon_{\mathrm{opn}} > \epsilon_{\mathrm{cls}}$ (*Figure 7a*).

The next step is to update the polarisation $\hat{\mathbf{p}}$ of the cells. In our description, the $\hat{\mathbf{p}}^\alpha$ for a given cell $\alpha$ is modeled to have a tendency to align with the orientation of its neighbours (ngb) without distinction between $\hat{\mathbf{p}}^{\mathrm{ngb}}$ and $-\hat{\mathbf{p}}^{\mathrm{ngb}}$. i.e., nematically. This is achieved with the following simple differential equation for $\hat{\mathbf{p}}_\alpha$ of any cell $\alpha$ in the interior:

$$\frac{d\hat{\mathbf{p}}_\alpha}{dt} = (\xi m_{\mathrm{align}} + \xi_s m_{\mathrm{stretch}} + \xi_{\mathrm{rand}}) \hat{\mathbf{p}}^\perp_\alpha, \tag{45}$$

where $\hat{\mathbf{p}}^\perp$ is such that

$$\hat{\mathbf{e}}_z = \hat{\mathbf{p}}_\alpha \times \hat{\mathbf{p}}^\perp_\alpha,$$

the $\hat{\mathbf{e}}_z$ is the vector normal to the plane of the paper.

$$m_{\mathrm{align}} = \hat{\mathbf{e}}_z \cdot \left[ \frac{1}{N_\alpha} \sum_{\beta \in \mathrm{ngb}} \mathrm{sign}(\hat{\mathbf{p}}_\alpha \cdot \hat{\mathbf{p}}_\beta)(\hat{\mathbf{p}}_\alpha \times \hat{\mathbf{p}}_\beta) \right]$$

is the rotational torque on $\hat{\mathbf{p}}_\alpha$ that aligns it with respect to the mean orientation of its neighbors and $\xi$ is the alignment rate.

$$m_{\mathrm{stretch}} = \hat{\mathbf{e}}_z \cdot [\mathrm{sign}(\hat{\mathbf{e}}_x \cdot \hat{\mathbf{p}}_\alpha)(\hat{\mathbf{p}}_\alpha \times \hat{\mathbf{e}}_x)]$$

is the rotational torque that aligns $\hat{\mathbf{p}}_\alpha$ along the orientation $\hat{\mathbf{e}}_x$ of the external uniaxial periodic stretching, when applied, and $\xi_s$ is the corresponding alignment rate.

The quantity $\mathrm{sign}(\hat{\mathbf{p}}_\alpha \cdot \hat{\mathbf{p}}_\beta)$ ensures that $\hat{\mathbf{p}}_\alpha$ nematically align with polarisation of the $N_\alpha$ neighbours in a mean sense. Similarly, the quantity $\mathrm{sign}(\hat{\mathbf{e}}_x \cdot \hat{\mathbf{p}}_\alpha)$ is to ensure that $\hat{\mathbf{p}}_\alpha$ aligns nematically along the orientation of external uniaxial stretch, $\hat{\mathbf{e}}_x$, when applied. The orientational noise $\xi_{\mathrm{rand}}$ is modeled as uncorrelated Gaussian noise of zero mean and strength $D_r$. This evolution rule also automatically ensures that the $\hat{\mathbf{p}} \cdot \hat{\mathbf{p}} = 1$ for all the cells. For any cells at the boundary, the only difference is that the alignment torque is

$$m_{\mathrm{align}} = \hat{\mathbf{e}}_z \cdot \mathrm{sign}(\hat{\mathbf{p}}_\alpha \cdot \hat{\mathbf{t}})(\hat{\mathbf{p}}_\alpha \times \hat{\mathbf{t}}),$$

where

$$\hat{\mathbf{t}} = \frac{\mathbf{r}_{\mathrm{next}} - \mathbf{r}_{\mathrm{before}}}{|\mathbf{r}_{\mathrm{next}} - \mathbf{r}_{\mathrm{before}}|},$$

and $\mathbf{r}_{\mathrm{next}}$ and $\mathbf{r}_{\mathrm{before}}$ correspond to the position vectors of the two contiguous cells of cell-$\alpha$ at the boundary.

In the case of colony elongation in the absence of external stretching, the initial condition for the positions $\{\mathbf{r}\}$ of the vertices and the polarisation $\{\hat{\mathbf{p}}\}$ of the cells is generated as follows. A confluent colony of 332 cells, a number similar to that observed experimentally, is initially confined within a circular region of radius $R_{\mathrm{colony}} \approx 320$ (see Appendix 2E for clarification regarding the simulation parameters.) The cells are provided with an initial polarisation $\hat{\mathbf{p}} = \cos\phi_p \hat{\mathbf{e}}_1 + \sin\phi_p \hat{\mathbf{e}}_2$ such that the orientation $\phi_p(x, y)$ is

$$\phi_p(x, y) = \frac{1}{2} \left( \tan^{-1} \left[ \frac{y - y_1}{x - x_1} \right] + \tan^{-1} \left[ \frac{y - y_2}{x - x_2} \right] \right) + \frac{\pi}{2}, \tag{46}$$

where $(x_1, y_1)$ and $(x_2, y_2)$ is the location of two $+1/2$ defects, which are taken to be symmetrically

separated by a distance of $1.6R_{\text{colony}}$ with respect to the center of the circle (also see Appendix 1D). An orientation shift of $\pi$ is randomly added to $\phi_p$ of individual cells such that the vector sum of $\hat{\mathbf{p}}$ over all the cells is zero. This step is irrelevant for the implementation of forces due to cell active stress and biased edge tensions due to their inherent nematic symmetry (*Equations 21 and 39*) but ensures that the total motile force from all the cells is zero. The cells are then provided with small active stress ($\sigma_a < 0$) and evolved under confinement for some time using the above rules till $\{\mathbf{r}\}$ and $\{\mathbf{p}\}$ are almost equilibrated. This configuration is then used as the initial condition for all the numerical experiments, in which the confinement is removed to allow the colonies to elongate (see *Video 8*). Different simulation conditions correspond to varying proportion of active cell stress ($\sigma_a < 0$), biased cell edge tension ($\lambda > 0$) and cell motility ($v_0$). The results from the simulations are then analysed in a similar manner as for the experimental data (*Figure 7—figure supplement 1*).

This set of polarity evolution rules for the interior and the boundary cells ensure that in the steady-state the orientation field $\phi_p$ satisfies in a coarse-grained sense, i.e., on longer length-scales, the Laplace equation $\nabla^2 \phi_p = 0$. The nematic field for the interior cells tends to have a uniform orientation, whereas the nematic field for the boundary cells tries to align along the colony boundary. Because of this boundary alignment tendency of the border cells, at least for the parameters used in our simulations (Appendix 2E), the total topological charge of the polarisation field is mostly conserved in our simulations, i.e., remains equal to the total initial charge of $+1$ (for a relevant study in another context see *van Bijnen et al., 2012*.) Due to the action of active cell stress (*Equation 21*) the cells have a tendency to elongate predominantly along $\hat{\mathbf{p}}$ thus creating a cell-shape nematic orientation field $\mathbf{q} = \cos\phi\hat{\mathbf{e}}_1 + \sin\phi\hat{\mathbf{e}}_2$ (also Appendix 1A-C.) Hence, the cell elongation anisotropy, as described earlier in Appendix 1, can loosely be thought of to be a readout of the internal polarisation of the cells. However, since the actual cell shape results from the interplay between cell mechanical properties and active cell stress, $\hat{\mathbf{p}}$ and $\mathbf{q}$ are not necessarily always fully aligned. Thus, for the cell-shape nematic field, the total topological charge can be less than $+1$ if the aligning tendency of internal cells dominates, e.g., when the cell active stress is large (see *Video 8*).

When the effects of external uniaxial stretching are included in the model, we take the $m_{stretch}$ term to dominate due to which the polarisation of the cells align along the orientation of uniaxial stretching $\pm\hat{\mathbf{e}}_x$. Hence, in this case, the role of initial and boundary conditions, as discussed above, is not quite important. As described in the section "Vertex model recapitulates symmetry breaking and shear decomposition" of the main paper, the competition between the cell active stress ($\sigma_a > 0$) and biased cell edge tension $\lambda > 0$ determines if the colony elongates either along the orientation of stretching or perpendicular to it (see *Video 9*).

## E. Parameters used and the non-dimensional groups present in the Vertex Model

Using reference scales for length ($l_0$), time ($t_0$) and energy ($U_0$), we get the following set of non-dimensional groups for the various parameters that are involved in the simulations. The non-dimensional counterparts of the parameters used in the simulations have been assigned an additional $'$ to make them distinct from the original.

1. Substrate friction: $\frac{\eta l_0^2}{t_0 U_0} \to \eta' = 1000$
2. Area modulus: $\frac{K l_0^4}{U_0} \to K' = 0.1$
3. Cell preferred area: $\frac{A_0}{l_0^2} \to A_0' = 1000$
4. Edge contractility: $\frac{\Lambda_0 l_0}{U_0} \to \Lambda_0' = 100$, $\frac{\Delta\Lambda l_0}{U_0} \to \Delta\Lambda = 20$, $\frac{\lambda l_0}{U_0} \to \lambda' \in [0, 50]$
5. Cellular active stress (negative sign indicates extensile): $\frac{\sigma_a l_0^2}{U_0} \to \sigma_a' \in [-8, 4]$.
6. Motility: $\frac{v_0 t_0}{l_0} \to v_0' = 0.05$
7. Polarisation alignment rate: $\xi t_0 \to \xi' = 0.03$
8. Stretch alignment rate: $\xi_s t_0 \to \xi' = 0.001$
9. Strength of Gaussian noise for polarisation: $t_0 D_r \to D_r' = 2.7 \times 10^{-6}$ (very small)
10. Relaxation time for edge tension fluctuations: $\frac{\tau}{t_0} \to \tau' = 10^3$
11. Total experimental time: $\frac{T_{\text{total}}}{t_0} \to T_{\text{total}}' = 4000$
12. T1 transition cutoffs: $\frac{\epsilon_{\text{opn}}}{l_0} \to \epsilon_{\text{opn}}' = 6$ and $\frac{\epsilon_{\text{cls}}}{l_0} \to \epsilon_{\text{cls}}' = 4$

13. Colony size: $\frac{R_{\text{colony}}}{l_0} \rightarrow R'_{\text{colony}} = 317$
14. Simulation time-step: $\frac{\Delta t}{t_0} \rightarrow \Delta t' = 1$

We can now choose the scales $l_0$, $t_0$, and $U_0$. We use the simulation time-step $\Delta t' = 1$. We get the values of total cumulative shear in the range $\approx 0.05 - 0.2$ that is comparable with the experimental values at the end of the 4000 simulation time. Since the total duration of the experiment is 2 hrs, this implies that $t_0 \approx 1.8$ s in experimental units. We also know from the experiments that the area of one cell is $\approx 400$ $\mu\text{m}^2$. Since, within our simulations, the preferred area of cells is taken as $A_0 = 1000$, this implies that $l_0 \approx 0.63$ $\mu\text{m}$.

The energy (or force) scale can be chosen arbitrarily since it is present in both the left hand and the right hand sides of *Equation 44*, and must only be chosen consistently. We now figure out the relative contributions to the forces that can come from different components for a cell: motile force, active force, basic edge contractility, active edge contractility, and isotropic force. For the values used the non-dimensional (simulation) values are:

1. cell motile force: $\approx \eta' v'_0 = 1000 \times [0, 0.05] = [0, 50]$
2. cell active force (extensile): $\approx |\sigma'_a \sqrt{A'_0}| = [0, 8] \times \sqrt{1000} \approx [0, 250]$
3. basic edge tension: $\Lambda'_0 = 100$
4. biased edge tension: $\approx \lambda' = [0, 45]$
5. cell area stress: $K' \sqrt{A'_0} \Delta A' \approx 3.16 \times [0, 15] \approx [0, 45]$

One can see that forces from all these sources are of the same order of magnitude. Specifically when we wish to compare the relative magnitudes of active edge contractility and the active stress, a quantity of relevance is $\frac{\lambda'}{\sigma'_a \sqrt{A'_0}}$. The values of the active terms $\sigma'_a$, $\lambda'$ and $v'_0$ used in the simulations are shown in *Figure 7—figure supplement 1*.

