## [Decision Letter]

**Acceptance summary:**

This is an interesting paper that gives useful and novel insight into collective cell mechanics. The study aims to understand the mechanism of tissue morphological changes using an in vitro cellular system, and examines the contribution of cellular events in the bulk in contrast to collective but singular events such as leader cell/finger emergence or topological defect dynamics. It provides additional evidence that confluent cell layers behave as active nematics, even if the individual cells are on average circular.

**Decision letter after peer review:**

Thank you for submitting your article "Epithelial colonies in vitro elongate through collective effects" for consideration by *eLife*. Your article has been reviewed by two peer reviewers, and the evaluation has been overseen by a Reviewing Editor and Aleksandra Walczak as the Senior Editor. The following individual involved in review of your submission has agreed to reveal their identity: Benoit Ladoux (Reviewer #2).

The reviewers have discussed the reviews with one another and the Reviewing Editor has drafted this decision to help you prepare a revised submission.

The study aims to understand the mechanism of tissue morphological changes using an in vitro cellular system, and examines the contribution of cellular events in the bulk in contrast to collective but singular events such as leader cell/finger emergence or topological defect dynamics. It provides additional evidence that confluent cell layers behave as active nematics, even if the individual cells are on average circular.

The study is interesting, however there are remaining issues that absolutely must be addressed before publication. Please do all the changes the reviewers ask for and reply to all their comments. The full reviews are included since it helps understand the reviewer's reasoning.

Reviewer #1:

This is an interesting paper which provides additional evidence that confluent cell layers are behaving as active nematics, even if the individual cells are circular on average. The experimental system is sufficiently simple that generic results appear to be possible. However there are gaps in the explanations which should be addressed.

First the authors measure the shape of an initially circular cell colony to understand whether and how it becomes anisotropic in shape. In the absence of external forcing the colony picks out a preferred direction at random and, after some initial fluctuations, the elongation along this direction increases with time. The anisotropic expansion is driven by fingers of cells and correlates strongly to the initial direction of nematic ordering within the colony. The main mesoscopic mechanism behind the expansion is cell elongation.

1) The paper states that the fingers appear at breaks in the acto-myosin cable that forms along the edge of the colony. They also say that the direction of expansion is correlated to the nematic order. So is there a correlation between the breaks and the initial nematic direction?

2) There is mention of topological defects inside the colony – although there cannot be too many of these in a system which is of order the size of the nematic correlation length. Do the defects affect the choice of the expansion direction? Is there a correlation between the positions of the fingers and the defects? This seems to be suggested in the modelling but not in the text describing the experiments. In particular I found the statement "Our analysis also showed that the cell orientation patterns are not homogeneous but spatially organized and directed by the presence of ±1⁄2 topological defects" unclear.

3) I wonder what would happen if the initial colony is larger so that there are initially regions of local nematic order in different directions?

4) It might also be interesting to look at a more detailed measure of the shape of colony e.g. length of interface relative to mean diameter.

The results are then compared to similar experiments in the presence of a cyclical stretch. The expansion is macroscopically similar, but the stretching direction defines the direction of expansion of the colony. External stretching helps fingers to grow. Now the expansion is primarily due to T1 transitions.

5) Does the stretching also line up the nematic ordering? If the fingers grow preferentially along the stretching direction, why doesn't the colony stretch more quickly compared to the non-stretched case?

Finally the authors obtain results for colonies with blocked cell-cell junctions. They find that these colonies still expand, but in a direction which is now perpendicular to the applied stretch for the stretched ones, and "in a non-isotropic manner" for the unstretched ones.

6) I did not understand "in a non-isotropic manner" in this context. Moreover, it seems surprising that the colonies expand at all and it would be helpful to include a discussion about this. Perhaps the modelling could help to interpret these results?

Reviewer #2:

This study aims to understand the mechanism of tissue morphological changes using an in vitro MDCK system, and examines the contribution of cellular events in the bulk (overall cell elongation, T1 changes, etc.) in contrast to collective but singular events such as leader cell/finger emergence or topological defect dynamics.

The strength of the manuscript is thorough quantification and having interesting modeling components. However, the experimental observations rest mainly at the correlational level. Mechanisms are being inferred from modeling, but there is minimal interaction between experiments and modeling. In addition, a few points should be clarified.

1) Authors show that a single cell elongate in perpendicular direction of the strain applied , however the collectives exhibit the elongation in the parallel direction to applied strain. This aspect of single versus collectives should be reasoned (specially for why collectives behave differently to external stimulus compared to single cells) and discussed further in their manuscript.

Could the model be used to understand why experimentally, single cells and E-cadherin-pertubed monolayer elongate perpendicularly to stretch direction, but a normal monolayer elongate in the direction of stretch?

2) The prediction of the model is that increasing polarized edge tension leads to T1 transition-dominant mechanism, while increasing active stress leads to cell-elongation-dominant mechanism. Can you do simple drug perturbation experiments to verify this?

3) Can the role of cell-substrate friction be probed in the model? Stress fibers were mentioned not to affect the elongation.

4) Also, since there is already a lot of literature in this field studying role of leader cells and even some on topological defect dynamics on colony expansion, can the authors do more analysis of their current data/ some experiments to elucidate the relative importance of these two singular events vs. those of the bulk cellular effects which was the focus of this study?

a) Are the paracellular actin cable breaks/defects at the periphery of the colony (which leads to leader cell emergence) distributed randomly or correlated to the direction of cell elongation?

b) If you induce leader cells by laser ablation of the periphery actin cables to induce leader cells in the direction perpendicular to colony elongation direction, will that impact strongly the original elongation direction?

c) Do the +1/2 topological defects dynamics correlate with the colony elongation direction and dynamics?

5) It would be interesting to recapitulate the main results with a different cell line to generalize the author findings.

6) The modeling should be further developed. In the present form, it mostly confirms the experimental results. (See for instance the remark on single versus collective dynamics).

7) The authors mentioned the existence of a supracellular actin cable (p4) but I do not see it on the data. If it is so, does the emergence of protrusive cells correspond to the rupture of the cable?

---

## [Author Response]

Reviewer #1:This is an interesting paper which provides additional evidence that confluent cell layers are behaving as active nematics, even if the individual cells are circular on average. The experimental system is sufficiently simple that generic results appear to be possible. However there are gaps in the explanations which should be addressed.First the authors measure the shape of an initially circular cell colony to understand whether and how it becomes anisotropic in shape. In the absence of external forcing the colony picks out a preferred direction at random and, after some initial fluctuations, the elongation along this direction increases with time. The anisotropic expansion is driven by fingers of cells and correlates strongly to the initial direction of nematic ordering within the colony. The main mesoscopic mechanism behind the expansion is cell elongation.1) The paper states that the fingers appear at breaks in the acto-myosin cable that forms along the edge of the colony. They also say that the direction of expansion is correlated to the nematic order. So is there a correlation between the breaks and the initial nematic direction?

We agree with the reviewer that since fingers appear at breaks in the acto-myosin cable and tissue expansion is correlated with the nematic order, it is worth to look for correlations of acto-myosin cable breaks and the nematic direction. We measured the mean nematic direction of MDCK colonies expressing mCherryLifeAct and GFP-myosin. We then located the positions of acto-myosin cable breaks and measured the angular distance between the center of the breaks and the mean nematic direction. In effect, there is no correlation between the breaks and the initial nematic direction.

The results are reported in the text and in a new Figure 5—figure supplement 4A and 4B:

“Moreover, we found no correlation between breaks of the acto-myosin cable surrounding the colony and the mean nematic direction (Figure 5—figure supplement 4), which suggests that breaks are uniformly distributed along the colony border.”

2) There is mention of topological defects inside the colony – although there cannot be too many of these in a system which is of order the size of the nematic correlation length. Do the defects affect the choice of the expansion direction? Is there a correlation between the positions of the fingers and the defects? This seems to be suggested in the modelling but not in the text describing the experiments. In particular I found the statement "Our analysis also showed that the cell orientation patterns are not homogeneous but spatially organized and directed by the presence of ±1⁄2 topological defects" unclear.

This is an interesting question. Indeed, in the beginning, we too had started our analysis in a similar direction. We first marked the location of future fingers on the colony circumference at t = 0 h by reverse tracing the fingers at t = 2 h. We then obtained the location of topological defects in the colony by using the procedure described in the paper. The mechanical influence of the +1/2 topological defect on the potential finger should be related to the distance of the defect from the circumference and the angle Δω between the defect and the finger along the colony circumference, where the angles are taken relative to the center of the colony. Hence, of all the +1/2 defects present in the colony at t = 0 h, the +1/2 defect that was connected to a particular finger was the one that (i) had the smallest angle Δω relative to the finger position and (ii) the distance of the defect from the center d_defect_ > R_colony_ /2, or half of the colony radius. The histogram of Δω_smallest_ for the fingers observed in N = 19 colonies used for this paper is shown in Author response image 1.

**Author response image 1. sa2fig1:** Distribution of the angular distance between fingers and their nearest defect.

As expected, the angles were skewed and more biased toward smaller values. There were many cases in which the position of the +1/2 defect and the location of the future finger at t = 0 was correlated as quantified by lower values of Δω. Despite this, there were cases that prevented us from formulating a strong statement about this correlation.

1) There were colonies when there was no +1/2 topological defect that could be clearly associated with a particular finger because:

The angle between the location of the defect and the finger with respect to the colony center was > 900, i.e. the defect is on the other side of the finger.The nearest topological defect in angle was much closer to the colony center than the defect, i.e. the actual influence of that particular defect on the finger was not quite clear.

2) The +1/2 topological defect that could be associated to a particular finger was not always stable, *i.e.* the defect that could be associated with a finger at t = 0 h disappears in intermediate steps.

3) There were also instances when the nematic field of cell shapes was strongly oriented in a particular direction which could be interpreted in terms of defects at the boundary or virtual defects outside the colony.

4) In some cases, the same topological defect was associated with more than one finger.

The orientation of the mean nematic field appears to be a simpler and more robust metric to evaluate the influence of the nematic field on the colony anisotropy. With this method we did not have to worry about any of the issues mentioned above. However, as described in the paper (Equation 1), we used the knowledge of the observed defect positions and strengths to recapitulate the orientation of the cell-shape nematic field θ using the expression that satisfies the Laplace equation ∇^2^θ = 0. Thus, we used the experimentally obtained defect information as a relevant readout for the actual nematic field – it indeed provides an excellent result as can be seen from Figure 5D. We have now stated these points more clearly in the paper.

“Moreover, the location of finger nucleation seemed to be biased towards the position of topological defects. However, some defect locations were not stable in time and in some cases, the nematic field of cell shapes could only be interpreted in terms of virtual defects outside the colonies, thus suggesting that the mean nematic direction is a better readout for the cell-shape nematic field.”

3) I wonder what would happen if the initial colony is larger so that there are initially regions of local nematic order in different directions?

We agree with the reviewer in his/her reasoning. Actually, we chose the initial size to match the correlation distance reported for circular MDCK colonies (Doxzen et al., 2013). To explore the effect of the initial size, we monitored the expansion of colonies of 750 µm in diameter. We observed that for colonies of this dimension, expansion was isotropic and symmetry was not broken. This experiment is now included in a new Figure 1—figure supplement 1 and discussed in the text.

“We observed that large colonies (750 µm in diameter) expanded isotropically (Figure 1—figure supplement 1). In contrast, colonies of 250 µm in diameter (Figure 1a), the typical coherence length of such epithelial tissues (Doxzen et al., 2013), expanded in a non-isotropic manner (Figure 1C).”

4) It might also be interesting to look at a more detailed measure of the shape of colony eg length of interface relative to mean diameter.

We acknowledge that other quantities may be interesting in order to quantify changes in shape for the colony. However, our focus is essentially on the symmetry breaking. In this context, we favor the usage of a nematic shape elongation tensor Q, and this choice will allow the comparison of elongations in vivo and in vitro, as well as contributions of single cells to the total shear. We propose to leave for future studies on shape transformation *per se* this length of interface over mean diameter readout which should indeed inform about shape complexity.

The results are then compared to similar experiments in the presence of a cyclical stretch. The expansion is macroscopically similar, but the stretching direction defines the direction of expansion of the colony. External stretching helps fingers to grow. Now the expansion is primarily due to T1 transitions.5) Does the stretching also line up the nematic ordering? If the fingers grow preferentially along the stretching direction, why doesn't the colony stretch more quickly compared to the non-stretched case?

We have evaluated the evolution of the mean nematic direction over time for stretching experiments and it is now included in the article (Figure 5—figure supplement 5 and in the text).

“Finally, when looking at the evolution of the mean cell elongation nematic field of colonies under uniaxial cyclic stretching, we observed that it did not change over time (Figure 5—figure supplement 5). The initial mean direction of cell elongation, either parallel or perpendicular to the external stretching, was maintained throughout 2 hours of external stretching. This suggests that average cell elongation alone does not determine colony elongation direction when subjected to uniaxial cyclic stretching. ”

As the reviewer points out, fingers grow preferentially along the stretching direction. However, we clarify in the following points why this do not lead to a quicker expansion in the stretching case:

– Fingers do not grow faster in the stretched case than in the control case (0,5 µm/min in both cases – Figure 4Dd and Figure 4—figure supplement 2F). Finger growth can therefore not lead to significant changes in the colony elongation rate.

– To show that fingers were affected by the direction of applied uniaxial cyclic stretching, we let the finger expand before applying stretching. This is distinct from our basic experiments, where fingers are yet to be formed when stretching is applied. This difference prevents comparisons between finger growth in Figure 4D and 4E.

These points are now clarified in the new version of the text and Figure 4—figure supplement 2F.

“We observed that, when growing perpendicular to the direction of force application, finger cells performed shorter displacements than when growing parallel to it. In the absence of externally applied cyclic uniaxial stretching, fingers grew a similar amount as when growing parallel the direction of applied uniaxial cyclic stretching and no bias was observed vis-à-vis the nucleation position (Figure 4D and Figure 4—figure supplement 2F).”

Finally the authors obtain results for colonies with blocked cell-cell junctions. They find that these colonies still expand, but in a direction which is now perpendicular to the applied stretch for the stretched ones, and "in a non-isotropic manner" for the unstretched ones.6) I did not understand "in a non-isotropic manner" in this context. Moreover, it seems surprising that the colonies expand at all and it would be helpful to include a discussion about this. Perhaps the modelling could help to interpret these results?

By “non-isotropic manner”, we meant that colonies still elongated anisotropically, along one preferred direction in the absence of application of cycling stretching. The quantification of this feature was captured by the Q_xx_ plots, which confirms this statement. This is clarified in the text:

“In the absence of externally applied uniaxial cyclic stretching, colonies treated with anti E-cadherin antibody expanded more than control colonies. Moreover, this expansion was still along one preferential direction (Figure 3D)”

We further developed the model to discuss and clarify the results for stretched colonies with blocked cell-cell junctions, see below point 1 of reviewer 2. Our new results confirmed the observed elongation.

Reviewer #2:This study aims to understand the mechanism of tissue morphological changes using an in vitro MDCK system, and examines the contribution of cellular events in the bulk (overall cell elongation, T1 changes, etc.) in contrast to collective but singular events such as leader cell/finger emergence or topological defect dynamics.The strength of the manuscript is thorough quantification and having interesting modeling components. However, the experimental observations rest mainly at the correlational level. Mechanisms are being inferred from modeling, but there is minimal interaction between experiments and modeling. In addition, a few points should be clarified.

We thank the reviewer for the positive evaluations.

To address these points, we tested specific inhibitors in experiments, *i.e.* colony response in the presence of ROCK inhibitor, and we extracted results which allow to identify causal relations in the phenomena. In addition, we developed further the model and ran new simulations to account for the stretching behavior of the colony as a collective of cells and as single cell behaviors. Experimental data reproduced the simulated colonies. These new results are reported below and they are included in the new version of the manuscript.

1) Authors show that a single cell elongate in perpendicular direction of the strain applied, however the collectives exhibit the elongation in the parallel direction to applied strain. This aspect of single versus collectives should be reasoned (specially for why collectives behave differently to external stimulus compared to single cells) and discussed further in their manuscript.Could the model be used to understand why experimentally, single cells and E-cadherin-pertubed monolayer elongate perpendicularly to stretch direction, but a normal monolayer elongate in the direction of stretch?

We thank the reviewer for asking this question.

We first note that in contrast to the control case, there is an inherent direction that is introduced by application of uniaxial cyclic stretching. Indeed, colonies subjected to uniaxial cyclic stretching elongated mainly due to T1 transitions (Figures 2 and 6E). However, when E-cadherin junctions were blocked, colonies elongated perpendicularly to the *x* direction *i.e.* along the *y* direction, similar to the direction observed for single cells (Figure 3). In our model, we did not explicitly include the effect of uniaxial cyclic stretching. Also, the model was meant to depict cell collectives as opposed to single cells. To address the questions posed by the reviewer, we modified the model and we performed new simulations. We can now use our model to understand the possible origins of these two distinct elongation behaviors.

As discussed in the paper, the colony elongation is mainly arising from T1 transitions and cell elongation. Based on the experimental observations discussed above, we propose that under the influence of stretching, cells actively “try” to elongate in the direction perpendicular to the external uniaxial stretching while undergoing neighbor exchanges in the direction parallel to the external uniaxial stretching. This effect could be incorporated in our model (Appendix 2A) by introducing an active cell stress of the formσactive=σa(e^xe^x−12I)where the choice σ_a_ > 0 would imply individual cells have a tendency to elongate along *y* (perpendicular to *x*). Similarly, for any edge with orientation ŝ, we use a biased edge tension distribution of the form (Appendix 2B)Λ= Λo+ λ ( 1−2(e^x⋅s^)2)where the choice λ> 0 would enhance colony elongation along *x* mainly due to T1 transitions.

We propose that the competition between the magnitudes of σ_a_ and λ will dictate the orientation of the colony elongation. More specifically, we propose that when E-cadherin junctions are strong, the λ term dominates, leading to colony elongation along *x* due to active T1 transitions. However, when E-cadherin levels are lower, we propose that the l term is smaller, leading to reduction in active T1 transitions and enhancement of single cell behavior. In this case, the term σ_a_ dominates over the term in λ, resulting in colony elongation perpendicular to *x*.

To implement this idea in our simulations, instead of directly introducing **ê**_*x*_ (direction of stretch) as discussed above, we indirectly include it through cell polarity, thus keeping the overall spirit of the model that is common to both stretching and control cases. We introduce an additional aligning term in the right hand side of Equation 45 of Appendix 2ξstretch e^z ⋅[sign (e^x ⋅ p^α)(p^α × e^x)]p^α⊥that tends to align the polarity of any cell α along *x* (control case corresponds to ξ_stretch_ = 0) to achieve the same effect as discussed above. The findings of these two cases (1) “stretching control” and (2) “stretching low polarized line tension” are shown in Video 9. They reproduce the distinct directions for the colony elongations.

Thus, based on combined insights provided from our experiments, analysis, and model, we propose that the competition between the strength of active T1 transitions and active cell stress dictate the overall elongation orientation. In the presence of external uniaxial cyclic stretching, when cell-cell junctions are strong, T1 transitions dominate, leading to elongation along *x*. On the other hand, when cell-cell junctions are weak (or absent), active cell stress dominates, thus leading to elongation along *y*.

These results are implemented in the Appendix 2, Video 9 and in the text and in Figure 7F-G :

“Our vertex model assumed that the cell elongation was the main readout for cell polarity, and it did not explicitly account for the effect of substrate stretching. […] When cell-cell junctions are weakened, active cell stress dominates, and colonies elongate perpendicular to the uniaxial stretching (Figure 3C and 3D), which could be thought of to be closer to a collection of single cells.”

2) The prediction of the model is that increasing polarized edge tension leads to T1 transition-dominant mechanism, while increasing active stress leads to cell-elongation-dominant mechanism. Can you do simple drug perturbation experiments to verify this?

We appreciate the positive evaluation for this prediction of our model. We also view this as an important insight in the behavior of tissues.

We performed a drug experiment as suggested by the reviewer to address this transition between T1 transition-dominant and cell-elongation-dominant mechanisms. We compared them in the new version of the manuscript. We found that colony response to external cyclic stretching involved ROCK activity, and that its inhibition led to a cell-elongation dominant mechanism, confirming that by modulating the strengths of active stress and polarized edge tension we change the dominant mechanism of colony elongation. This is now reported in the text and in new Figure 8 and Figure 8—figure supplement 1.

“Stretching-dependent elongation is mediated by ROCK. We showed that upon stretching, cells reduced their speed and myosin structures appeared (Figure 3E and 3F). […] Strikingly, the application of a ROCK inhibitor leads to single cell elongation dominating over T1 (orange dot in Figure 8c), effectively suppressing the effect of uniaxial cyclic stretching on the mode of colony deformation (Figure 8—figure supplement 1).”

3) Can the role of cell-substrate friction be probed in the model? Stress fibers were mentioned not to affect the elongation.

This is an interesting question. Substrate friction is taken into account in the model through the termη dridton the left-hand side of Equation 44 of Appendix 2, but was not probed in the model explicitly. However, it is not explicitly connected to stress fibers. If the underlying assumption of the question is that cell-substrate interaction would induce stress fibers, we propose the following analysis. We did not see major stress fibers in the control case, and this suggests that stress fibers *per se* were not essential for elongation of the colony. However, we did see and report stress fibers in the stretching case in Figure 3F as reinforcements between cells mediated by cyclic stretch. Tentatively, we could link this appearance to increased T1 transitions, and this would support an absence for the role of stress fibers in elongation as noted by the reviewer. We would prefer not to lengthen the Manuscript with this conjecture, but would be open if the reviewer thinks that some additional clarification is needed.

4) Also, since there is already a lot of literature in this field studying role of leader cells and even some on topological defect dynamics on colony expansion, can the authors do more analysis of their current data/ some experiments to elucidate the relative importance of these two singular events vs. those of the bulk cellular effects which was the focus of this study?a) Are the paracellular actin cable breaks/defects at the periphery of the colony (which leads to leader cell emergence) distributed randomly or correlated to the direction of cell elongation?

Along the question #1 of reviewer 1, we analyzed the locations of the breaks at the periphery, and we find no correlation between the nematic field and the breaks position. The results are reported in the text and in a new Figure 5—figure supplement 4:

“Moreover, we found no correlation between breaks of the acto-myosin cable surrounding the colony and the mean nematic direction (Figure 5—figure supplement 4), which suggests that breaks are uniformly distributed along the colony border.”

b) If you induce leader cells by laser ablation of the periphery actin cables to induce leader cells in the direction perpendicular to colony elongation direction, will that impact strongly the original elongation direction?

Following the reviewer’s suggestion, we triggered breakage of the cable by local injection of cytochalasin D. Although it induced the disruption of the cable and after disruption the cable “healed”, no growth of finger was observed. We include a supplementary figure (Figure 5—figure supplement 1), a video (Video 5) and the results are now inserted in the main text.

“We therefore explored the possibility of inducing the growth of fingers and therefore set the direction of elongation of the colonies. Breakage of the actomyosin cable by laser ablation induces the appearance of leader cells (Reffay et al., 2014). Hence we attempted to trigger the growth of fingers by locally injecting cytochalasin D using a micropipette. The transient injection of this actin polymerization inhibitor was followed by the disruption of the acto-myosin cable (Video 5 and Figure 5—figure supplement 1). However, the cable reformed, and fingers did not appear. This result shows that breakage of the cable alone doesn’t trigger the growth of fingers in our colonies, and suggests that other mechanisms may be involved.”

c) Do the +1/2 topological defects dynamics correlate with the colony elongation direction and dynamics?

This question is related to question #2 of reviewer #1.

In order to determine any possible correlation between topological defects and finger position direction, we systematically measured the angles and distances between fingers and the closest defect (see Author response image 1). Although we found a bias in the angular distribution, there were cases which prevented us from reporting this correlation (see the response to question 2 of the first reviewer). Briefly, we observed that for some colonies defect locations were not stable and, in some cases, the nematic field of cell shapes could be interpreted in terms of defects outside the colony. In contrast, nematic direction was more robust as this readout appeared to be a more reliable predictor for the direction of colony expansion. This is now discussed in the text.

“Moreover, the location of finger nucleation seemed to be biased towards the position of topological defects. However, some defect locations were not stable in time and in some cases, the nematic field of cell shapes could only be interpreted in terms of virtual defects outside the colonies, thus suggesting that the mean nematic direction is a better readout for the cell-shape nematic field.”

5) It would be interesting to recapitulate the main results with a different cell line to generalize the author findings.

We agree with the reviewer and we now include experiments using Caco-2 cells and MCF 10A cells. Experiments confirmed our results: spontaneous elongation of the colony was observed for the 3 epithelial cell lines and nematic field matching direction of elongation could be measured for MDCK and MCF 10A cells (it could not be measured for Caco-2 cells because the cell boundaries were not clear enough to obtain reliable nematic fields).

The results are discussed in the text and in Figure 1—figure supplement 2, and Figure 5—figure supplement 2 (also changes in Materials and methods were inserted):

“In addition, we explored if other epithelial cell lines would behave in a similar manner. Circular epithelial colonies of human epithelial colorectal adenocarcinoma cells (Caco2) and human mammary epithelial cells (MCF-10A) also elongated along the main axis of elongation and by the same magnitude that MDCK cells (Figure 1—figure supplement 2). We note that elongation observed during this time for the three epithelial cell lines was similar in magnitude to tissue elongation observed during in vivo morphogenesis, for instance in the wing blade in *Drosophila* (Etournay et al., 2015).”

“We followed the evolution of the cell elongation nematic field in different MDCK and MCF 10A colonies during expansion. We first obtained the spatio-temporal cell elongation nematic orientation field ϕ(x,y,t) (see Materials and methods) on the experimental time-lapse images (see Figure 5B, Figure 5—figure supplement 2AC, Video 6 and Appendix 1C)”

6) The modeling should be further developed. In the present form, it mostly confirms the experimental results. (See for instance the remark on single versus collective dynamics).

We developed further the model as suggested by the review (see the answer to question 1 above) to account for cell stretching. By doing so, we retrieved the response of colonies to stretching in control case and with cell-cell contacts weakened.

Also, the reviewer pointed out that the model predicted that “increasing polarized edge tension leads to T1 transition-dominant mechanism, while increasing active stress leads to cell-elongation-dominant mechanism.” We explored this shift by the new experiments with ROCK inhibition without and with stretching.

These successful interactions between model and experiments are now included in the new version of the text and in the Discussion.

“We perturbed active stress and line edge tension by inhibiting ROCK, which has been reported to be involved in cell-cell contact integrity in vivo (Nishimura and Takeichi, 2008; Ewald et al., 2012), and recently, in cell responses to stretching in vitro (Hart et al., 2020). This led to experimentally blocking the ability of colonies to respond to the externally applied uniaxial cyclic stretching. By doing so, colonies which primarily elongate through cell intercalations, shifted to a single cell elongation driven mechanism.”

7) The authors mentioned the existence of a supracellular actin cable (p4) but I do not see it on the data. If it is so, does the emergence of protrusive cells correspond to the rupture of the cable?

We had included the acto-myosin cable (now in Figure 4—figure supplement 1). Moreover, we are now adding analysis of the local breakage mentioned by the reviewer (Figure 5—figure supplement 1).